# Cones 2: Customizable Image Synthesis with Multiple Subjects

**Zhiheng Liu**[1*†]    **Yifei Zhang**[2*†]    **Yujun Shen**[3]    **Kecheng Zheng**[3]    **Kai Zhu**[1,4]
**Ruili Feng**[1,4]    **Yu Liu**[4]    **Deli Zhao**[4]    **Jingren Zhou**[4]    **Yang Cao**[1‡]

[1]USTC        [2]SJTU        [3]Ant Group        [4]Alibaba Group

## Abstract

Synthesizing images with user-specified subjects has received growing attention due to its practical applications. Despite the recent success in single subject customization, existing algorithms suffer from high training cost and low success rate along with increased number of subjects. Towards controllable image synthesis with multiple subjects as the constraints, this work studies how to efficiently represent a particular subject as well as how to appropriately compose different subjects. We find that the text embedding regarding the subject token already serves as a simple yet effective representation that supports arbitrary combinations without any model tuning. Through learning a residual on top of the base embedding, we manage to robustly shift the raw subject to the customized subject given various text conditions. We then propose to employ layout, a very abstract and easy-to-obtain prior, as the spatial guidance for subject arrangement. By rectifying the activations in the cross-attention map, the layout appoints and separates the location of different subjects in the image, significantly alleviating the interference across them. Both qualitative and quantitative experimental results demonstrate our superiority over state-of-the-art alternatives under a variety of settings for multi-subject customization. Project page can be found here.

## 1 Introduction

The remarkable achievements of text-to-image generation models [1–14], have garnered widespread attention due to their ability to generate high-quality and diverse images. To allow synthesizing images with user-specified subjects, customized generation techniques [15–17] propose to fine-tune the pre-trained models on a few subject-specific images. Despite the notable success in single subject customization [15–20], multi-subject customization remains seldom explored but better aligns with the practical demands in real life.

Recent studies [16, 17] have investigated multi-subject customization through joint training, which tunes the model with all subjects of interest simultaneously. Such a strategy has two drawbacks. First, they require learning separate models for each subject combination, which may suffer from exponential growth when the number of subjects increase. For example, the customization of objects $\{A, B, C\}$ fails to inherit the knowledge obtained from the customization of objects $\{A, B\}$. Second, different subjects may interfere with each other, causing the issues that some subjects fail to show up in the final synthesis or the subject attribute gets confused among subjects (*e.g.*, a cat with the features of another dog). This phenomenon is particularly evident when the semantic similarity between subjects is high (see Fig. 4).

---

[*]Equal contribution.
[†]Work performed during internship at Alibaba DAMO Academy.
[‡]Corresponding author.

37th Conference on Neural Information Processing Systems (NeurIPS 2023).

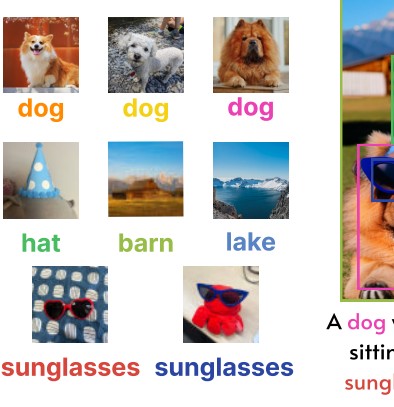 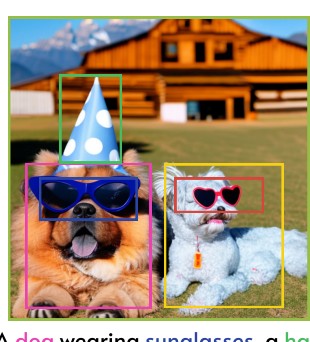 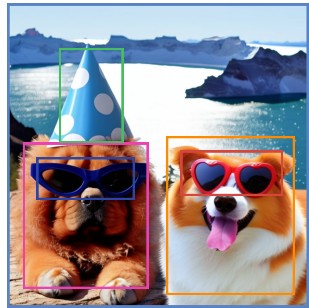

Figure 1: **Customizable image generation** with the subjects listed on the left. **Cones 2** is highlighted from three aspects. (1) Using a simple yet effective representation to register a subject, we can compose various subjects arbitrarily *without any model tuning*. (2) Employing spatial layout, which is very easy to obtain in practice, as a guidance, we can *control the specific location* of each subject and meanwhile *alleviate the interference* across subjects. (3) Our method achieves appealing performance even under some *challenging settings*, such as customizing the synthesis with six or more subjects and exchanging the sunglasses on the two dogs.

In this work, we present Cones 2, a novel approach for multi-subject customization using a pre-trained text-to-image diffusion model. Our method utilizes a simple yet effective representation to register a subject and enables the arbitrary composition of various subjects without requiring any model retraining. To that end, we decompose the challenging task of multi-subject customization into two components: how to efficiently represent a subject and how to effectively combine different subjects. Given a set of subjects and their photos (3-5 for each), our goal is first to bind the characteristic of each specific subject to a "plugin" that can be used flexibly. Driven by this, we fine-tune the text encoder part of a pre-trained text-to-image diffusion model with images of a specific subject, making the tuned model can customize this specific subject. Moreover, we propose a text-embedding-preservation loss, which limits the output of the tuned text encoder to only differ from the original text encoder in token embedding regarding the specific subject. Then we calculate the mean difference between the tuned text encoder with the original text encoder to derive the residual token embedding which can robustly shift the raw category to the customized subject (*e.g.*, dog → customized dog).

To effectively combine different subjects, we propose a layout guidance method to control the generation process. More formally, we employ pre-defined layout, a very abstract and easy-to-obtain prior, to guide different subjects to show up in different positions by rectifying the activation in the cross-attention maps. We encourage all subjects to show up in the final synthesis by strengthening the activations of the target subject. Simultaneously, to prevent the subject attribute gets confused, we weaken the activations of the irrelevant subjects. In addition, to make this easy to implement in practice, we define the layout as a set of subject bounding boxes with subject annotation, which describes the spatial composition of the customized subjects and is easy for users to specify in advance. Through our method, users can compose various subjects arbitrarily with a pre-defined layout (see Fig. 1).

Our method is evaluated under a variety of settings for multi-subject customization involving extensive subject categories such as pets, scenes, decorations, *etc.* Qualitative and quantitative results demonstrate that, compared to existing baselines, our method exhibits competitive performance in terms of both text alignment to input prompt and visual similarity to the target images. It is noteworthy that our method even facilitates the customization of a larger number of subjects (*e.g.*, six subjects in Fig. 1), which is a far more challenging setting in practice.

## 2 Related work

**Large-scale text-conditioned image synthesis.** Synthesizing images from the language description has received growing attention due to its ability to generate high-quality and diverse images. Earlier

works [21] explored the utilization of language description into GAN as a condition on specific domains under the closed-world assumption. With the development of diffusion models [22, 23] and large-scale multi-modality models [24], text-conditioned image synthesis has shown remarkable improvement in an open-vocabulary text description. Specifically, GLIDE [3], DALLE2 [4], StableDiffusion [5] and Imagen [6] are representative diffusion models that can produce photorealistic outputs. Autoregressive models such as DALLE [1], Make-A-Scene [2], CogView [25] and Parti [26] have also shown exciting results. Although these models demonstrate an unparalleled ability to synthesize images, they require time-consuming iterative processes to achieve high-quality image sampling. Recent large text-to-image GAN models such as StyleGAN-T [7], GALIP [27], and GigaGAN [8] also demonstrated unprecedented semantic generation, which is orders of magnitude faster when sampling.

**Customized image generation.** Thanks to the significant progress of large-scale text-to-image models, users can adopt these well-trained models to generate customized images with user-specified subjects. There are two earliest attempts to solve the customized generation through few-shot images of one specific subject, *i.e.* Text Inversion [18] and DreamBooth [15]. Concretely, Text Inversion [18] represents a new subject by learning an extra identifier word and adding this word to the dictionary of the text encoder. DreamBooth [15] binds rare new words with specific subjects through few-shot fine-tuning the whole Imagen [6] model. To compose multiple new concepts together, Custom [16] chooses to only optimize the parameters of the cross-attention in the StableDiffusion [5] model to represent new concepts and then joint trains for the combination of multiple concepts. In addition, Cones [17] associates customized subjects with activating a small cluster of neurons in the diffusion model. Although both Custom [16] and Cones [17] have explored combination multi-subject customization, they suffer from high training costs and low success rates along with the increased number of subjects. In this work, we study how to efficiently represent a particular subject as well as how to appropriately compose different subjects. Specifically, learning a residual on top of the base embedding can represent a new concept, and the introduction of layout into the attention map can help the model generate more accurate user-specified subjects. We find these design choices lead to better results in directly composing different subjects than joint training.

**Spatial guidance in diffusion models.** To further enhance the controllability of synthesizing images, some works [28–32] have tried to explore how to guide the generation process by more spatial information. Composer [28] directly adds spatial information as a condition input during the training phase. ControlNet [29] and T2I-Adapters [30] add spatial information to the pre-trained model by training a new adapter. Prompt-to-prompt [31] presents a training-free edit method by editing the cross-attention. In addition, a diffusion-based image translation [32] keeps the generated spatial structure by limiting the cross-attention map. Inspired by these works, we also adopt the layout as the spatial guidance for subject arrangement that can well appoint and separate the location of different subjects in the image, significantly alleviating the interference across them.

## 3 Method

Given a set of subjects and their photos (3-5 for each) from different views, we aim to generate new images of any combination containing those subjects vividly and precisely. We accomplish this by combining subject-specific residual token embeddings with a pre-trained diffusion model and guiding the generation process with a layout. The overall framework is presented in Fig. 2. Specifically, we represent each subject as a residual token embedding shifted from its base category. Adding the residual token embedding to the base category embedding can yield the corresponding subject in the generated images. We present how to get this residual token embedding in Sec. 3.2. At inference time, subjects failing to show up and the subject attribute getting confused among subjects are two key problems in multi-subject customized generation. To address these issues, we present a method of composing subjects by leveraging layout guidance in Sec. 3.3.

### 3.1 Text-conditioned diffusion model

Diffusion models learn a data distribution by the gradual denoising of a variable sampled from a Gaussian distribution. This corresponds to learning the reverse process of a fixed-length Markov chain. In text-to-image tasks, the training objective of a conditional diffusion model $\epsilon_\theta$ can be simplified as a reconstruction loss,

$$L_{\text{rec}} = \mathbb{E}_{\mathbf{x},\mathbf{c},\boldsymbol{\epsilon}\sim\mathcal{N}(0,1),t}[\|\boldsymbol{\epsilon}_\theta(\mathbf{x}_t, E(\mathbf{c}), t) - \boldsymbol{\epsilon}\|_2^2], \tag{1}$$

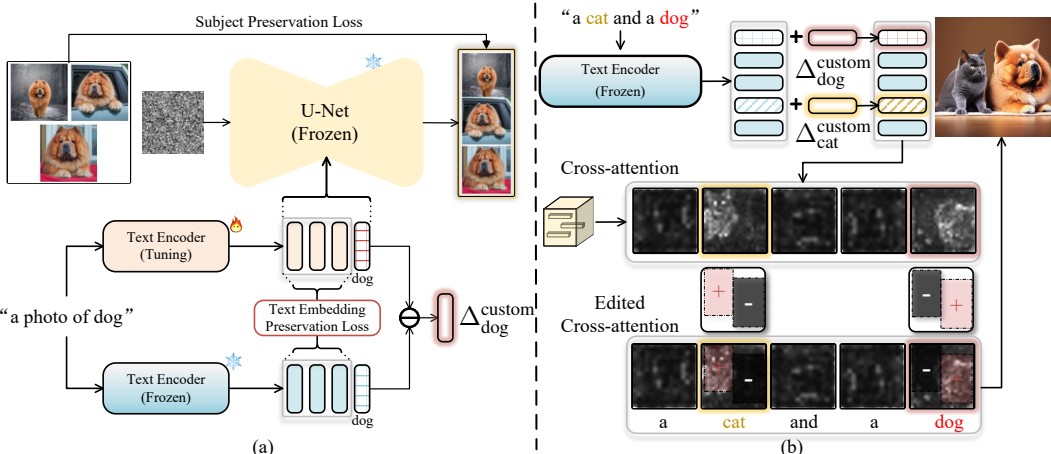

Figure 2: **Illustration of the proposed approach.** (a) We first learn a residual token embedding (*e.g.*, $\Delta_{\text{dog}}^{\text{custom}}$) on top of the base embedding to register a user-specified subject, which allows composing various subjects arbitrarily without further model tuning. (b) Given a layout as the spatial guidance, we then arrange the subjects by rectifying the activations in cross-attention maps, which enables the control of the location of each subject and reduces the interference between them.

where $t \sim \mathcal{U}([0, 1])$ is the time variable, $E$ is a pre-trained text encoder and $\mathbf{x}_t = \alpha_t \mathbf{x} + \sigma_t \boldsymbol{\epsilon}$ is a noised image from the ground-truth image $\mathbf{x}$. The parameters $\alpha_t$ and $\sigma_t$ are coefficients formulating the forward diffusion process. The model $\boldsymbol{\epsilon}_\theta$ is conditioned on the text embedding $E(\mathbf{c})$ and $t$. The text embedding $E(\mathbf{c})$ is injected into the model $\boldsymbol{\epsilon}_\theta$ through the cross-attention mechanism. At inference time, the network $\boldsymbol{\epsilon}_\theta$ is sampled by iteratively denoising $\mathbf{x}_T \sim \mathcal{N}(0, \mathbf{I})$ using either deterministic samplers [33–35] or stochastic sampler [23].

### 3.2 Representing subjects with residual token embedding

**Representing subjects with residual text embedding.** Our goal is first to represent each subject with a residual text embedding among the output domain of the text encoder. An ideal residual text embedding $\Delta^{\text{custom}}$ that can robustly shift the raw category to a specific subject. For example, the model $\boldsymbol{\epsilon}_\theta$ with embedding input $(E(\text{"a photo of dog"}) + \Delta_{\text{dog}}^{\text{custom}})$ can truly generate a photo of specific "dog". One way to get this objective is to calculate an embedding direction vector [32] from a source (original) text encoder to the target (fine-tuned) text encoder $E^{\text{custom}}$. The fine-tuned text encoder $E^{\text{custom}}$ needs to be able to customize subject $s$ combined with the original diffusion model $\boldsymbol{\epsilon}_\theta$. Similarly as DreamBooth [15], $E^{\text{custom}}$ can be trained with the subject-preservation loss, as

$$L_{\text{sub}}(E^{\text{custom}}) = \mathbb{E}_{(\mathbf{x},\mathbf{c}) \sim D_s, \boldsymbol{\epsilon} \sim \mathcal{N}(0,1), t}[\|\boldsymbol{\epsilon}_\theta(\mathbf{x}_t, E^{\text{custom}}(\mathbf{c}), t) - \boldsymbol{\epsilon}\|_2^2], \qquad (2)$$

where $D_s = \{(\mathbf{x}_j^s, \text{"a photo of } s\text{"}) | \mathbf{x}_j^s \in X^s\}$ is the reference few-shot data of subject $s$.

**Regularization with a text-embedding-preservation loss.** The residual text embedding obtained according to the previous section can only perform single-subject customized generation. Since those residual text embeddings are applied to the entire text, any two of them can admit significant conflicts so that they cannot be combined together directly while carrying out inference. Therefore, we propose a text-embedding-preservation loss to make the residual text embedding mainly act on the text embedding regarding the subject token. The core idea is to minimize the difference between $E^{\text{custom}}$ and $E$ for tokens apart from the subject token $s$. Take the "dog" case above as an example, we sample 1,000 sentences $C_{\text{dog}} = \{\mathbf{c}^i\}_{i=1}^{1000}$ containing the word "dog" using ChatGPT [36], like "a dog on the beach", and then minimize the difference between $E^{\text{custom}}$ and $E$ for all the token besides "dog". In detail, given any caption (*e.g.* $\mathbf{c} = $ "a dog on the beach"), we split its text embedding into a sequence $(E(\mathbf{c}) = (E(\mathbf{c})_a, E(\mathbf{c})_{\text{dog}}, \cdots, E(\mathbf{c})_{\text{beach}}))$. Then we wish $\|E(\mathbf{c})_p - E_p^{\text{custom}(\mathbf{c})}\|_2^2 = 0$ for any $p$ that is not equal to "dog". Namely, the text-embedding-preservation loss is a regularization, as

$$L_{\text{reg}}(E^{\text{custom}}) = \mathbb{E}_{\mathbf{c} \sim C_{\text{dog}}}[\sum_{p \in \mathbf{c}, p \neq s} \|E^{\text{custom}}(\mathbf{c})_p - E(\text{c})_p\|_2^2], \qquad (3)$$

**Algorithm 1** N-Subject Customization with Layout Guidance

---

**Require:** Prompt $\mathbf{c}$, customized set $S = \{s_i\}_{i=1}^N \subset \mathbf{c}$, pre-trained residual token embeddings $\{\Delta_{s_i}^{\text{custom}}\}_{i=1}^N$, guidance layout $\mathbf{M} = \{\mathbf{M}_s : s \in S\}$.
1: Edit the text embedding: $E^{\text{final}}(\mathbf{c}) = E(\mathbf{c}) \oplus \{\Delta_{s_i}^{\text{custom}}\}_{i=1}^N$;
2: Define guidance layout for each $s \in S$ from $\mathbf{M}$:

$$\mathbf{M}_s(i,j) = \begin{cases} \gamma^+ & (i,j) \in R_s^{\text{show}} \\ \gamma^- & (i,j) \in R_s^{\text{irrelevant}} \\ 0 & \text{Otherwise} \end{cases}$$

3: Sample $\mathbf{x_T} \sim \mathcal{N}(0, \mathbf{I})$;
4: **for** $t = T, T-1, \ldots, 1$ **do**
5:     $\mathbf{CA} \leftarrow \epsilon_\theta(\mathbf{x}_t, E^{\text{final}}(\mathbf{c}), t)$;
6:     $\mathbf{CA}_{\text{edited}} \leftarrow \text{EditedCA}(\mathbf{CA}, \mathbf{M}, \mathbf{c})$;
7:     $\mathbf{x}_{t-1} \leftarrow \epsilon_\theta(\mathbf{x}_t, E^{\text{final}}(\mathbf{c}), t)\{\mathbf{CA}_{\text{edited}}\}$.
8: **end for**

---

where $p$ traverses all tokens inside sentence $\mathbf{c}$ except the subject token $s$. Our complete training objective then comes as

$$L = L_{\text{sub}} + \lambda L_{\text{reg}}, \tag{4}$$

where $\lambda$ controls for the relative weight of the text-embedding-preservation term. As shown in Fig. 2a, after the customized text encoder is obtained, we derive the *residual token embedding* of "dog" via computing the average shift of $E^{\text{custom}}(\mathbf{c})_{\text{dog}}$ over these 1,000 sentences from $E(\mathbf{c})_{\text{dog}}$, as

$$\Delta_{\text{dog}}^{\text{custom}} = \frac{1}{|C_{\text{dog}}|} \cdot \sum_{\mathbf{c} \in C_{\text{dog}}} (E^{\text{custom}}(\mathbf{c})_{\text{dog}} - E(\mathbf{c})_{\text{dog}}). \tag{5}$$

**Inference with residual token embedding.** The residual token embedding we get aforementioned can be used directly in any subject combinations involving them without further tuning. As shown in Fig. 2b, when we do customized generation with $N$ specific subjects $s_1, s_2, \cdots, s_N$, all we need is to fetch the pre-computed $\Delta_{s_1}^{\text{custom}}, \Delta_{s_2}^{\text{custom}}, \cdots, \Delta_{s_N}^{\text{custom}}$ and add them to the token embedding, as

$$E^{\text{final}}(\mathbf{c})_{s_i} = E(\mathbf{c})_{s_i} + \Delta_{s_i}^{\text{custom}}, i = 1 \cdots, N. \tag{6}$$

In fact, the operation in Eq. (6) is all in the token dimension. This characteristic endows our method with significant convenience and high efficiency for large-scale applications. On the one hand, any pre-trained residual token embedding $\Delta_i^{custom}$ can be used repeatedly and combined with another $\Delta_j^{custom}$. On the other hand, for each subject, we merely need to store a float32 vector, getting rid of storing large parameters as in previous methods [15–17].

### 3.3 Composing subjects with layout guidance

The text-to-image diffusion models [4–6] commonly inject the text embedding $E(\mathbf{c})$ to its diffusion model $\epsilon_\theta$ via the cross-attention mechanism. The attention map among a cross-attention layer is $\mathbf{CA} = (\mathbf{W}_Q \cdot \varphi(\mathbf{x}_t)) \cdot (\mathbf{W}_K \cdot E(\mathbf{c}))$, where $\varphi(\mathbf{x}_t)$ denotes the transformed image feature and $\mathbf{W}_Q, \mathbf{W}_K$ denotes the parameters for computing query and key. The cross-attention map directly affects the spatial layout of the final generation [31]. Below we will discuss how to improve the quality of customized generation by rectifying the activations in the cross-attention map.

**Strengthening the signal of target subject.** One issue in multi-subject customization is that some subjects may fail to show up. We argue that this is caused by insufficient activations in the cross-attention map of these subjects. To avoid this, we choose to strengthen the signal of the target subject in the region where we want it to show up.

**Weakening the signal of irrelevant subject.** Another issue in multi-subject customization is that the subject attribute gets confused among subjects, *i.e.* the subjects in generated images may contain characteristics from the other subjects. We argue that this is due to the overlapping activation regions of different subjects in the cross-attention map. To avoid this, we choose to weaken the signal of each subject appearing in the region of the other subjects.

**Layout-guided iterative generation process.** Combining the above two ideas, we present a method to guide the generation process according to a pre-defined layout $\mathbf{M}$. In practice, we define the layout $\mathbf{M}$ as a set of subject bounding boxes and then get the guidance layout $M_s$ for each subject $s$. In detail, as shown in Fig. 2b we divide $M_s$ into different regions: we set the value of $M_s$ to a positive value $\gamma^+ \in \mathbb{R}^+$ in the region where we want the subject $s$ to show up (denote as $R_s^{\text{show}}$) and set the value of $M_s$ to a negative value $\gamma^- \in \mathbb{R}^-$ in the region that is irrelevant to the subject $s$ (denote as $R_s^{\text{irrelevant}}$). At the inference time, we replace all the output of cross-attention with edited results at every generation step, as

$$\text{EditedCA}(\mathbf{CA}, \mathbf{M}, \mathbf{c}) = \text{Softmax}(\mathbf{CA} \oplus \{\eta(t) \cdot \mathbf{M}_{s_i} | i = 1, \cdots, N\}) \cdot (\mathbf{W}_V \cdot E(\mathbf{c})), \quad (7)$$

where $\oplus$ denotes the operation that adds the corresponding dimension of $\mathbf{CA}$ and $\mathbf{M}$, which is also visualized in Fig. 2b and $\eta(t)$ is a concave function controlling the edit intensity at different time $t$. The implementation details refer to Algorithm 1.

# 4 Experiments

## 4.1 Experimental setups

**Datasets.** For fair and unbiased evaluation, we select subjects from previous papers [15, 18, 16, 17] spanning various categories for a total of 15 customized subjects. It consists of two scenes, five pets and eight objects. We perform extensive experiments on various combinations of subjects, explaining the superiority of our approach.

**Evaluation metrics.** We evaluate our approach with two following metrics for customized generation proposed in Textual Inversion [18]. (1) Image similarity, which measures the visual similarity between the generated images and the target subjects. For multi-subject generation, we calculate the image similarity of the generated images and each target subject separately and finally calculate the mean value. (2) Textual similarity, which evaluates the average CLIP [24] similarity between all generated images and their textual prompts. To this end, we use a variety of prompts with different settings to generate images, including modifying the scenes, attributes, and relation between subjects.

**Baselines.** To evaluate our generation quality, we compare our approach with three state-of-art competitors, *i.e.*, *DreamBooth* [15] that fine-tunes all parameters in diffusion model; *Custom diffusion* [16] that optimizes the newly added word embedding in text encoder and a few parameters in diffusion model, namely the key and value mapping from text to latent features in the cross-attention; and *Cones* [17] that finds a small cluster of neurons in diffusion model corresponding each customized subject. As Custom diffusion and Cones demonstrated, we omit Textual Inversion [18] as it performs much less competitively. And the implementation details of our approach and those of baselines are also reported in the Appendix A.

## 4.2 Main results

In this section, to demonstrate the superiority of our approach, we conduct experiments on authentic images from diverse categories, including objects, pets, backgrounds, *etc.*. We not only present the qualitative results between our approach and other baselines but also showcase quantitative comparison. This further substantiates the effectiveness of our approach.

**Qualitative comparison.** As depicted in Fig. 3, we present a collection of generated images featuring two to four subjects. For single-subject generation, as shown in Appendix B.1, our approach achieves comparable results to competing methods while requiring significantly less storage space. However, as the number of subjects increases, the other three methods fail to include certain subjects and exhibit attribute confusion, resulting in generated images that deviate from the reference images. In contrast, our approach consistently produces highly visually accurate images for all subjects. It is important to note that our approach utilizes learned single-subject residual token embeddings for seamless combinations without retraining, thereby avoiding exponential training costs associated with the other methods. The next section will discuss this in detail.

**Quantitative comparison.** In the context of generating customized subjects with varying numbers, we have carefully selected four evaluation metrics: textual similarity, visual similarity, required storage space, and computational complexity. As shown in Tab. 1, for single-subject generation, our approach exhibits slightly lower visual and textual similarity compared to DreamBooth while

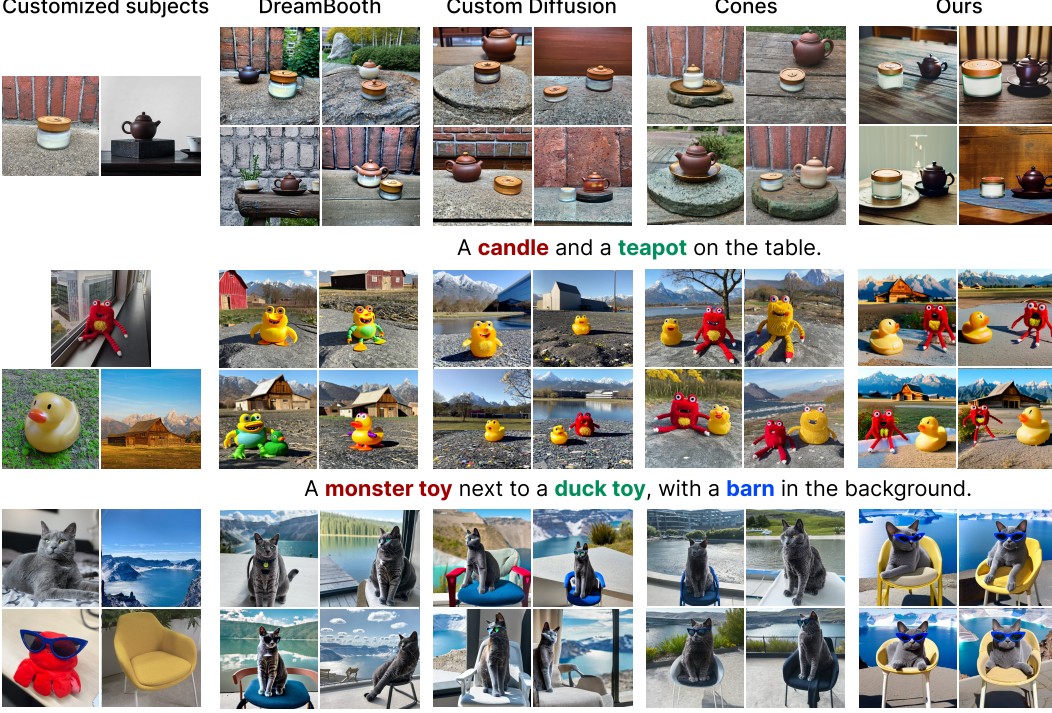

Customized subjects     DreamBooth     Custom Diffusion     Cones     Ours

A **candle** and a **teapot** on the table.

A **monster toy** next to a **duck toy**, with a **barn** in the background.

A **cat** is wearing **sunglasses** and sitting on a **chair**, with a **lake** in the background.

Figure 3: **Qualitative comparison** of multi-subject generation ability between our approach and baselines. Our approach surpasses existing alternatives with higher success rate in generating these subjects and less attribute confusion among different subjects.

remaining comparable to the other two methods. However, as the number of subjects increases, our approach consistently outperforms the other methods across all four evaluation metrics. This demonstrates its effectiveness in capturing subject characteristics, maintaining fidelity to the given prompt, and achieving higher efficiency in practical applications. Please note that the storage space and computational complexity presented in Tab. 1 for our approach assume that there are no existing learned single-subject residual token embeddings. However, in practice, if we already have required subject-specific residual token embeddings for multi-subject generation, no additional storage space or computational complexity is needed. This ability to seamlessly combine existing models without retraining is a unique advantage of our approach. In contrast, other methods require new storage space and training time for generating multiple new subjects.

**User study.** We conduct a user study to further evaluate our approach. The study investigates the performance of the four methods on the multi-object customization task. For each task and each method, we generated 80 images from 4 subjects combination, 4 conditional prompts, and 5 random seeds, resulting in 1,280 generated images for the whole user study. We presented two sets of questions to participants to evaluate image similarity and textual similarity. Taking the prompt "A cat and a dog on the beach" as an example, where "cat" and "dog" are customized subjects, we provided reference images of the customized subjects and asked participants questions like: "Does the image contain the customized cat?" to evaluate visual similarity. For textual similarity, based on the textual description, we presented questions like "Does the image contain a cat and a dog?". As shown in Tab. 2, our approach is most preferred by users regarding both image and text alignment.

### 4.3 Towards challenging cases

In this section, we further illustrate our superiority by showcasing two scenarios: generating a larger number of customized subjects and generating subjects with high semantic similarity that other methods fail to achieve.

Table 1: **Quantitative comparisons**. Our approach outperforms other methods in all aspects of multi-subject customization, particularly in three-subject and four-subject generation. The complexity metric is determined by calculating the number of fine-tuning iterations required for each method to generate a certain combination of $n$ subjects.

| | Method | Text Alignment | Image Alignment | Storage | Complexity |
|---|---|---|---|---|---|
| **Single Subject** | DreamBooth [15] | 0.314 | **0.727** | 3.3 GB | $O(n)$ |
| | Custom Diffusion [16] | 0.327 | 0.721 | 72 MB | $O(n)$ |
| | Cones [17] | **0.331** | 0.722 | $(1.43 \pm 0.34)$ MB | $O(n)$ |
| | Ours | 0.330 | 0.725 | 4.8 KB | $O(n)$ |
| **Two Subjects** | DreamBooth [15] | 0.278 | 0.664 | 3.3 GB | $O(n^2)$ |
| | Custom Diffusion [16] | 0.284 | 0.676 | 72 MB | $O(n^2)$ |
| | Cones [17] | 0.292 | 0.685 | $(3.41 \pm 0.56)$ MB | $O(n^2)$ |
| | Ours | **0.309** | **0.708** | 9.6 KB | $O(n)$ |
| **Three Subjects** | DreamBooth [15] | 0.252 | 0.649 | 3.3 GB | $O(n^3)$ |
| | Custom Diffusion [16] | 0.270 | 0.658 | 72 MB | $O(n^3)$ |
| | Cones [17] | 0.281 | 0.663 | $(4.96 \pm 0.70)$ MB | $O(n^3)$ |
| | Ours | **0.304** | **0.689** | 14.4 KB | $O(n)$ |
| **Four Subjects** | DreamBooth [15] | 0.241 | 0.604 | 3.3 GB | $O(n^4)$ |
| | Custom Diffusion [16] | 0.254 | 0.623 | 72 MB | $O(n^4)$ |
| | Cones [17] | 0.271 | 0.638 | $(7.75 \pm 0.56)$ MB | $O(n^4)$ |
| | Ours | **0.299** | **0.673** | 19.2 KB | $O(n)$ |

Table 2: **User study.** The value represents the percentage of users that score positive for the image generated corresponding to the given questions. The results show that our approach is the most preferred by users for multi-subject customization, on both image and text alignment.

| | DreamBooth [15] | | Custom Diffusion [16] | | Cones [17] | | Ours | |
|---|---|---|---|---|---|---|---|---|
| | Text Alignment | Image Alignment | Text Alignment | Image Alignment | Text Alignment | Image Alignment | Text Alignment | Image Alignment |
| **Single Subject** | 71.35% | **71.50**% | **76.85**% | 67.60% | 76.85% | 69.60% | 75.05% | 69.05% |
| **Two Subjects** | 52.58% | 43.13% | 59.88% | 46.83% | 62.55% | 57.50% | **77.87**% | **69.75**% |
| **Three Subjects** | 57.78% | 31.83% | 58.20% | 34.28% | 64.87% | 37.94% | **79.20**% | **64.42**% |
| **Four Subjects** | 36.42% | 25.63% | 40.73% | 25.44% | 42.10% | 28.75% | **77.35**% | **59.08**% |

**Customization with similar subjects.** To illustrate the superiority of our approach in mitigating attribute confusion among multiple customized subjects, we select Cones, one of the methods that perform well in multi-subject generation among the three baselines, and compare the generated images with our approach in challenging cases shown in Fig. 4. We observe that when the raw categories of the customized subjects have high semantic similarities, especially in the case of two customized dogs, Cones exhibits a notable propensity for attribute confusion to arise. In contrast, our approach demonstrates excellent performance in both visual and textual similarities.

**Customization with a large number of subjects.** As shown in Fig. 3, we observe a significant decrease in the quality of generated images by other methods as the number of customized subjects increases. However, our approach shows a relatively smaller impact. Therefore, in Fig. 5, we present the generated images with an increased number of customized subjects, further demonstrating the effectiveness of our approach.

## 4.4 Ablation studies

**Verify the effect of strength and weaken cross-attention map.** We conduct ablation experiments to examine the individual effects of strengthening the target subject region and weakening irrelevant subject regions as shown in Fig. 6. We observe that strengthening the target subject alone can lead to attribute confusion between subjects. For instance, it may result in a customized subject exhibiting attributes of another subject. However, when only irrelevant subjects are weakened, certain subjects may fail to show up or exhibit a lack of specific attributes. Furthermore, for simple combinations like "mug + teapot," satisfactory results could be achieved with 30 steps of guidance. However, for more challenging combinations such as "cat + dog," 50 steps of guidance were required to achieve better attribute binding results.

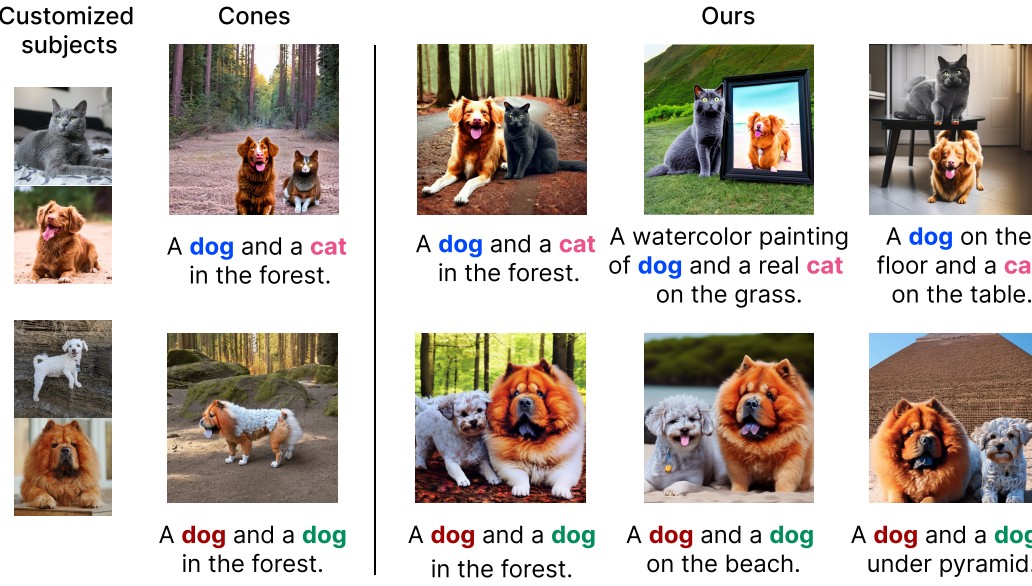

Figure 4: **Visualizations of customization with challenging cases.** When the subjects that need to be customized belong to the category with high semantic similarity (shown in the first row) or even the same category (shown in the second row), the baseline using joint training has a serious attribute confusion problem, while our approach circumvents this problem.

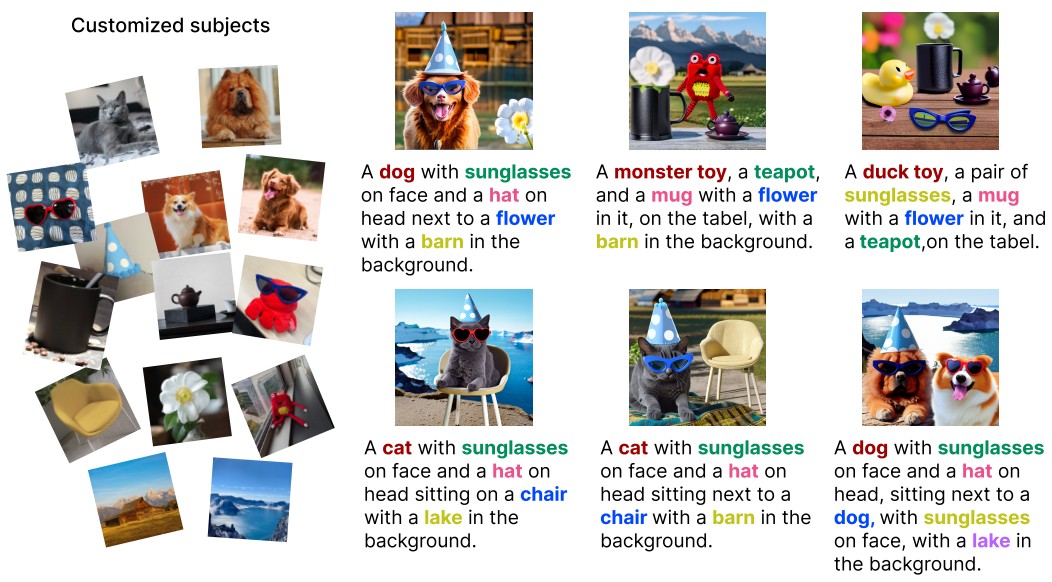

Figure 5: **Visualizations of customization with a large number of subjects.** Here we show diverse generation results of customizing 5 and 6 subjects.

**Verify the effect of guidance in generation process.** Recent studies [37, 38] have also demonstrated that incorporating guidance during the sampling process leads to superior generation results. We select *Cones* [17], which exhibits relatively better performance among competing methods, as our baseline, and compare it with the state-of-the-art semantic guidance approach as well as our guidance approach. As shown in Fig. 7, combining the Attend and Excite [38] improves the generation quality compared to cones, and further improvement is achieved when combine with our guidance approach. However, overall, our approach outperforms others, showcasing the best performance. This proves that the problem of attribute confusion in other methods can't be solved by simply adding a guidance algorithm while combining our residual token embedding with our guidance algorithm can solve it.

| Strengthen only | Weaken only | T=30 | T=50 |
|:---:|:---:|:---:|:---:|

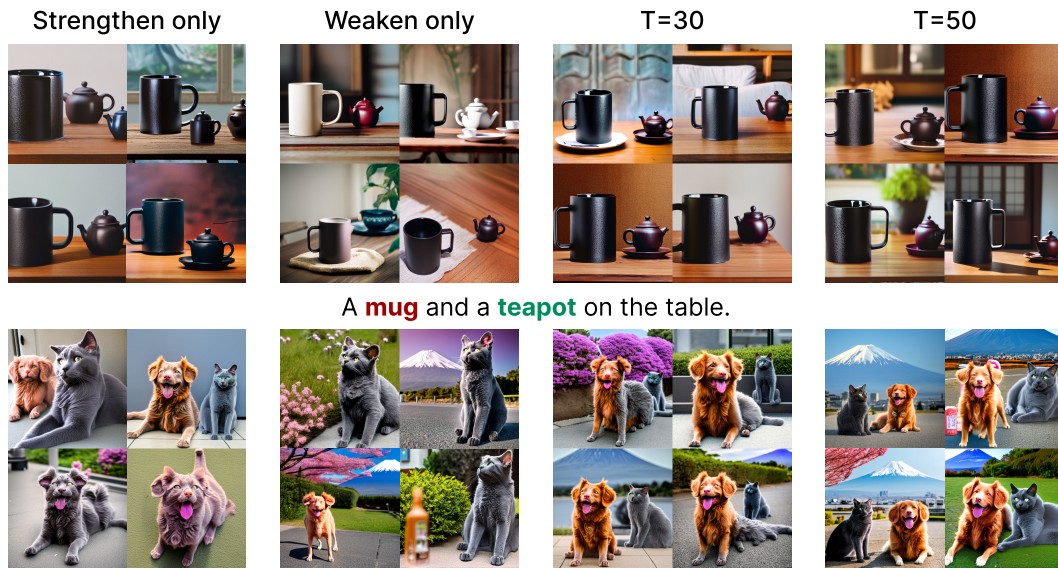

A **mug** and a **teapot** on the table.

A **dog** and a **cat** under mount fuji.

Figure 6: **Ablation study on the effectiveness of strengthening and weakening cross-attention map.** The first column shows that only strengthening leads to attribute confusion. The second column shows only weakening leads to some subjects failing to show up. The last two column shows that those challenging cases require longer guidance.

| Customized subjects | Cones | Cones *w/* Att | Cones *w/* Layout | Ours |
|:---:|:---:|:---:|:---:|:---:|

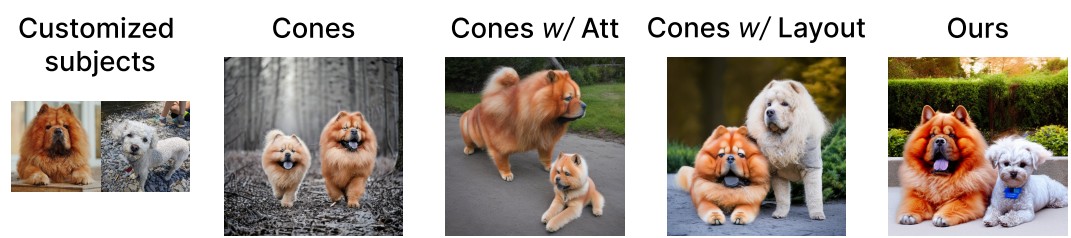

A **dog** and a **dog** in the garden on a sunny day.

Figure 7: **Qualitative comparison for ablations** on Attend-and-Excite [38] and our layout guidance. "Att" refers to Attend-and Excite and "Layout" refers to our guidance method. These results show that the baseline method obtained through joint training cannot avoid attribute confusion by simply combining a certain guidance method. Correspondingly, our method can solve this problem.

## 5   Conclusion

This paper proposes a novel approach for multi-subject customization. Our method combines subject-specific residual token embeddings with a pre-trained diffusion model and utilizes easy-to-obtain layout prior to guiding the generation process. This allows us to combine individually learned subject-specific residual token embeddings for multi-subject customization without retraining. Our method consistently delivers exceptional performance even in challenging scenarios, including the customization of image synthesis with six subjects and the customization of semantically similar subjects. Through qualitative and quantitative experiments, we demonstrate our superiority over existing state-of-the-art methods in various settings of multi-subject customization. These results highlight the effectiveness and robustness of our method.

## Acknowledgments

This work is supported by National Key R&D Program of China under Grant 2020AAA0105701 and Alibaba Group through Alibaba Research Intern Program.

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

# Appendix

## A  Experimental details

We supplement the experimental details of each baseline method and our method in this section. For better generation quality, we use Stable Diffusion v2-1-base [1] as the pre-trained model. For a fair comparison, we use 50 steps of DDIM [33] sampler with a scale of 7.5 for all the above methods. All experiments are conducted using one A-100 GPU.

**Textual Inversion [18].** We use the third-party implementation of huggingface [39] for Textual Inversion. We train each subject-specific token with the recommended[2] batch size of 4 and a learning rate of 0.002 for 3000 steps. In particular, we initialize the subject-specific token with the corresponding class token. For example, to customize a specific cat, we initialize the subject-specific token "<cat>" with the original "cat" token.

**DreamBooth [15].** We use the third-party implementation of huggingface [39] for DreamBooth. Training is with a batch size of 2, learning rate $5 \times 10^{-5}$, and training steps of $800 \times$ number of subjects. **Custom Diffusion [16].** We use the official implementation[3] for Custom Diffusion. Training is with a batch size of 2, learning rate $1 \times 10^{-5}$ and training steps of $250 \times$ number of subjects.

**Cones [17].** We use the official implementation for Cones. The batch size of the training stage is set to 2, the learning rate is $4 \times 10^{-5}$ and training steps is $1200 \times$ number of subjects.

**Ours.** For our approach, We train each subject-specific residual token embedding with a batch size of 1 and a learning rate of $1 \times 10^{-6}$ for 3,000 steps. At inference time, the layouts are appointed by bounding boxes given by the users to indicate the location of each subject. We use a positive value of $+2.5$ to strengthen the signal of the target subject and we use a negative value of $-1 \times 10^{-5}$ to weaken the signal of irrelevant subjects. Furthermore, we guide all $50$ steps with the layout guidance in the whole generation process to get good customized generation results.

**User study.** For two- to four-subject generation tasks, we design four different subject combinations for each task. This will yield 12 subject combinations in total. For each subject combination, we design four different text prompts to generate images with 5 random seeds. We conduct this procedure to all four methods. With such settings, each method generates 80 different images for each task. We give each generated image 4-8 questions for testing image alignment (2-4 questions) and text alignment (2-4 questions). The number of questions is proportional to the number of subjects used to customize the image (average 6 questions per generated image). Finally, we shuffle the order of all the image-question pairs and assigned them to 25 different users for scoring, and finally summarized the results. In detail, every user needs to score $4 \times 4 \times 5 \times 6 = 480$ questions for each task and for each method.

## B  More results

In this section, we present additional visualization results that provide a comprehensive demonstration of the superiority of our proposed method.

### B.1  More comparisons with other baselines

We conduct further comparison between our approach and three other baselines. As shown in Fig. A1, regarding the generation of single subjects, the four methods exhibited similar performance. However, when dealing with semantically similar subjects, such as a dog and a cat, as well as scenarios involving three or more subjects, our approach clearly exhibit superior performance. Moreover, as shown in Fig. A2, we further showcase additional generated results, providing further evidence of the robustness of our method.

---

[1]https://huggingface.co/stabilityai/stable-diffusion-2-1
[2]https://github.com/rinongal/textual_inversion
[3]https://github.com/adobe-research/custom-diffusion

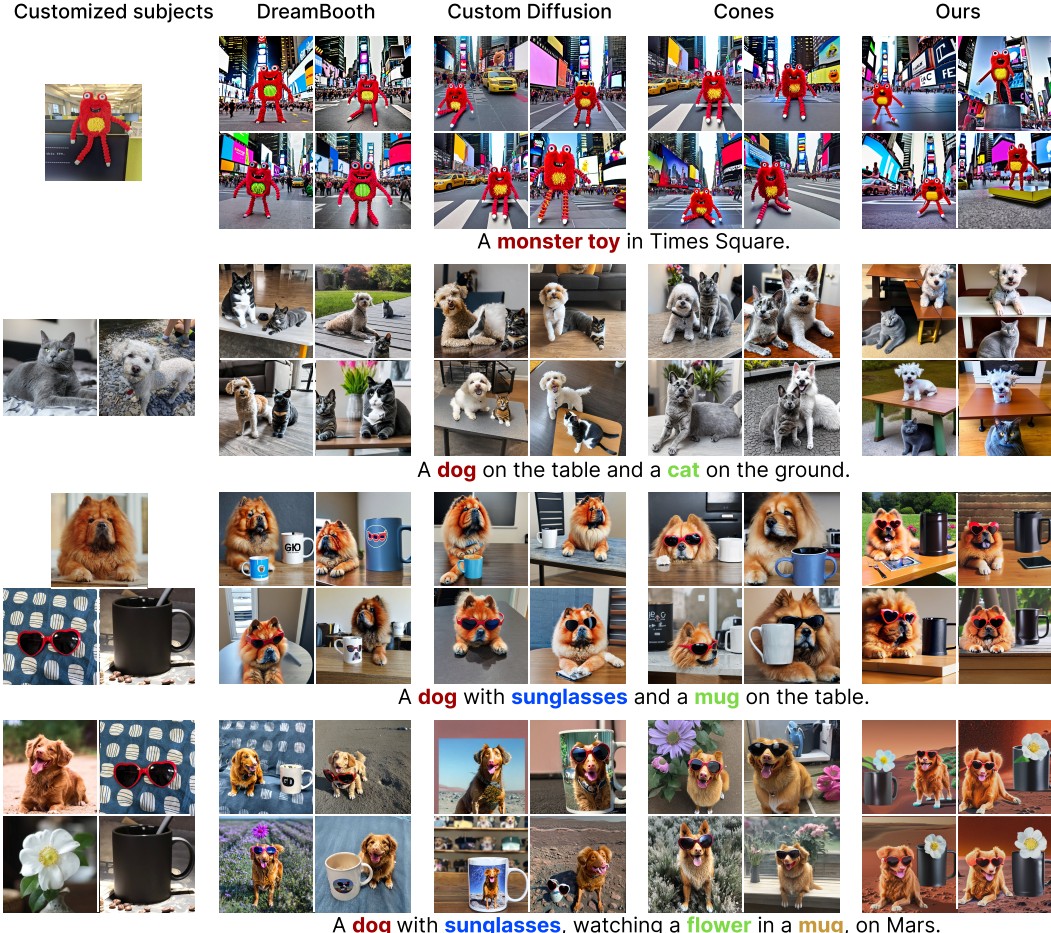

Figure A1: More comparisons of our approach and existing baselines.

## B.2 More challenging cases

As shown in Fig. A3, we present a larger number of images generated by our approach, featuring a greater diversity of customized subjects. In comparison with other methods, we observe that when the number of customized subjects reaches four, the performance of other methods significantly deteriorates. In contrast, our approach can generate a larger number of customized subjects, exemplifying the superiority.

## B.3 Comparison with image composition method

we select Paint by example [40] as a baseline of image composition methods. In order to compare the effect of the baseline and our method more intuitively, we use Paint by example to inpaint the image generated by our method with the reference image as input. We conduct visualization results in Fig. A4. Generated results of our method have better visual similarity, and we will provide more comparisons in the revision. Besides, We consider multi-subject customized generation is different with image composition in two a aspects. The purpose of multi-subject customized generation is to implant all user-provided subjects into the diffusion model; so that the model can generate various images of all subjects vividly guided by prompts. However, the purpose of image compositing is to insert an object into another image in a realistic way, without guidance from prompts. In addition, to compose multiple subjects by the image composition methods, multiple iterations of inference are required, especially when there is an overlapping relationship among subjects.

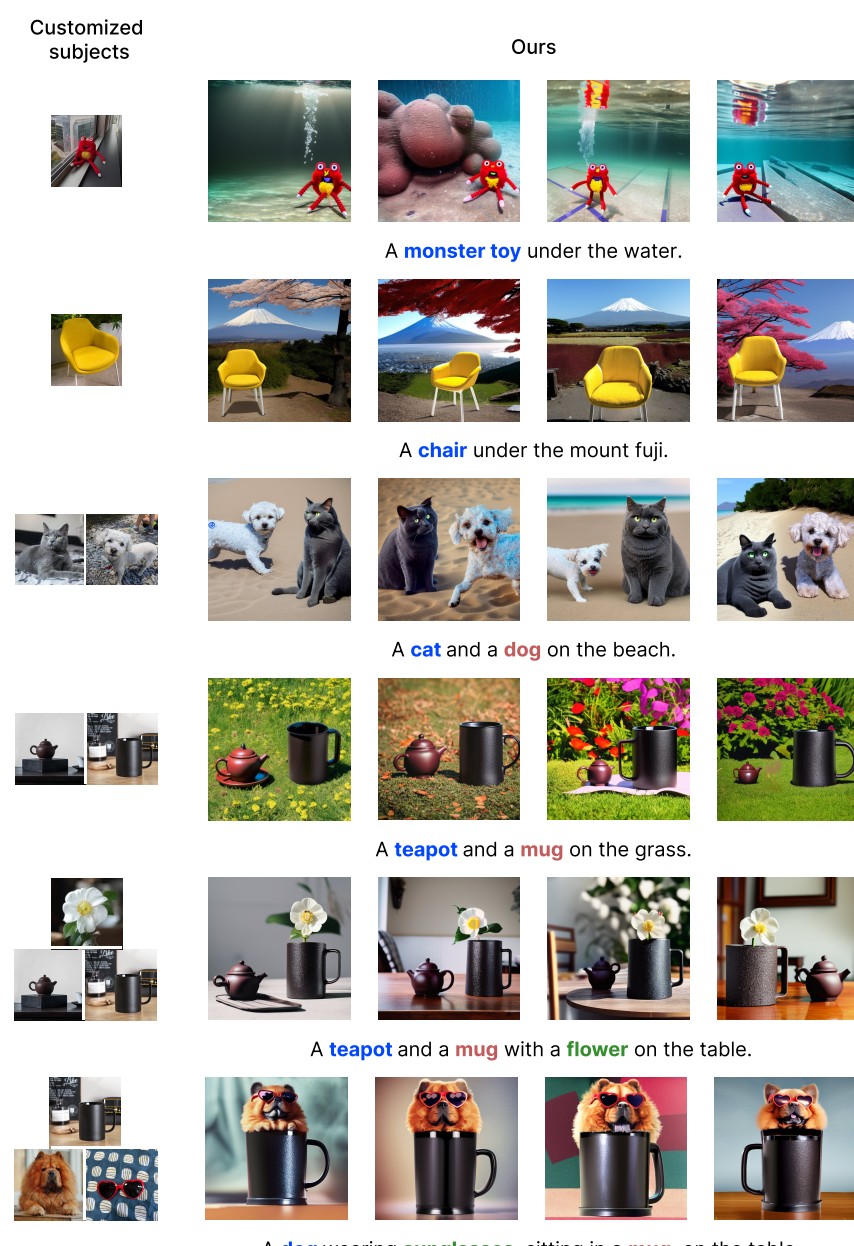

Figure A2: More results of our approach.

## B.4    More fine-grained control

Actually, users have the flexibility to input a more specific mask, enabling them to achieve fine-grained control. Experimental results in Fig. A5 demonstrate that our method can use specific mask for sampling.

## B.5    About generated results with more subjects

As for more subjects, it is limited by the capabilities of the pre-trained model itself. As shown in Fig. A6, more involved subjects usually decrease the final visual quality. we observe significantly more failure cases when generating five or more subjects. Several recent works [17, 16, 38] point out that stable diffusion is a struggle in generating multiple subjects. A potential way to ease this issue may be to apply our method to a better text-conditioned diffusion model.

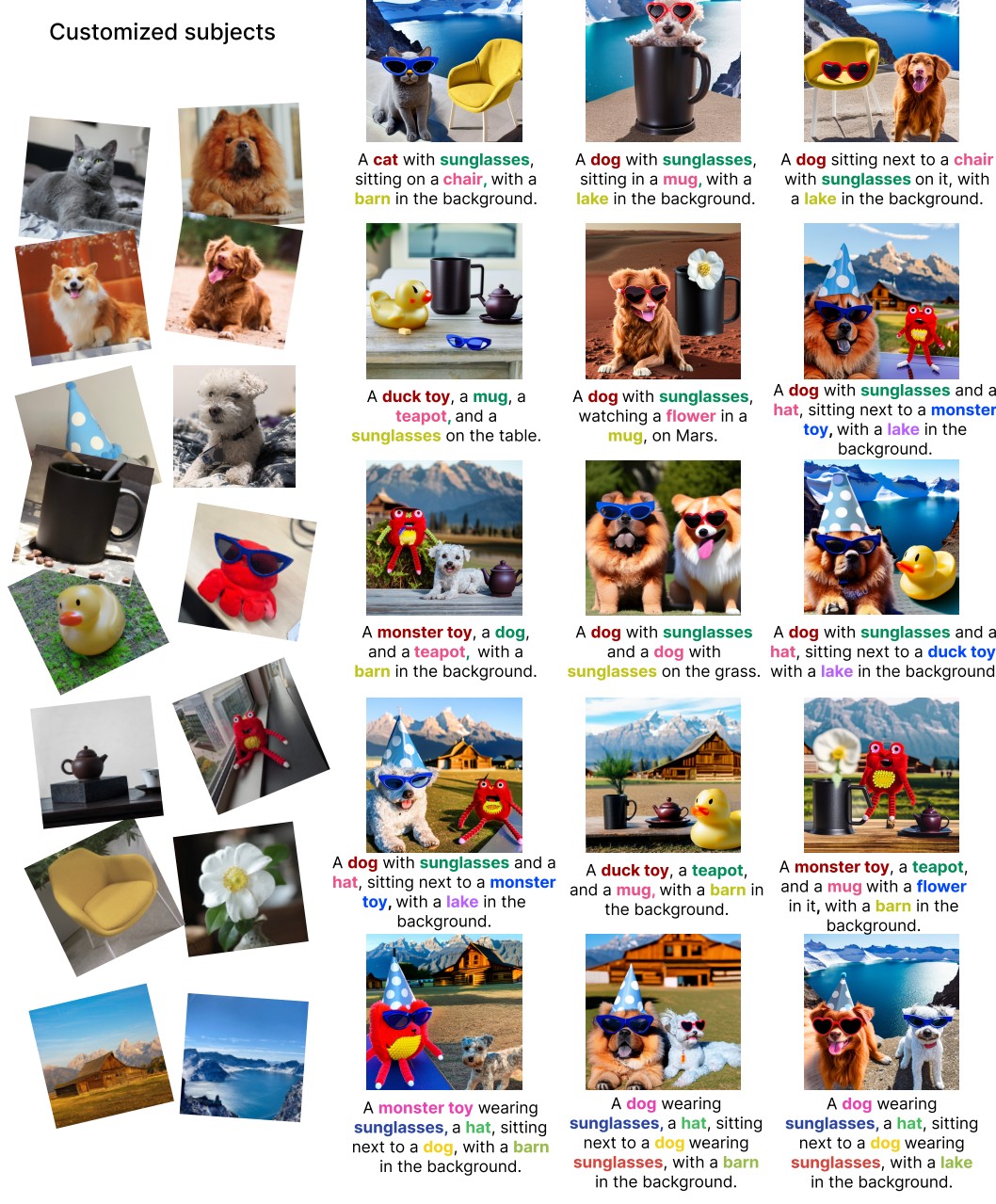

Figure A3: More results of multi-subject generation.

## B.6 Comparison with fast customized methods

Currently, some works [41, 42] focus on achieving faster customized generation by pretraining an image encoder and using it to encode a reference image during generation. However, these methods share some common issues. They require collecting data in advance to train the encoder, which can limit their generalization. We compare our method with Elite in Fig. A7. Compared to methods that fine-tune for each individual subject, these approaches may exhibit difficulty in preserving fine details. In addition, these methods can only encode one image at a time, they cannot customize multiple subjects simultaneously.

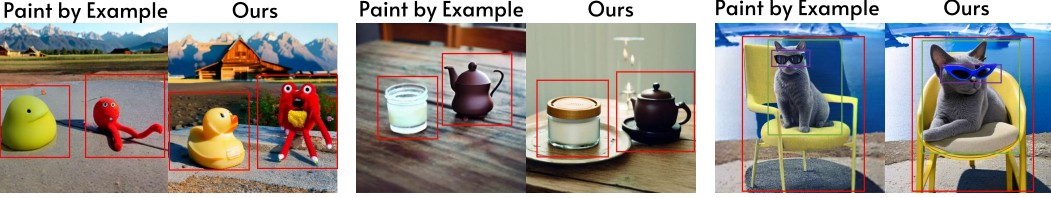

Figure A4: Comparisons with Paint by Example.

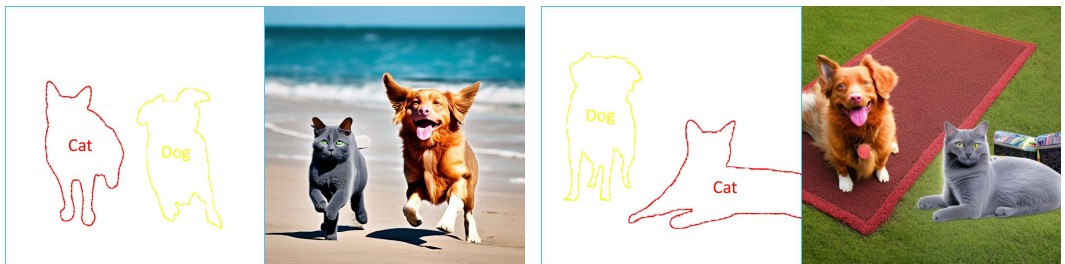

A dog and a cat on the beach.        A dog and a cat on the grass.

Figure A5: Generated results using specific mask. We use a more specific mask as a prior to get more fine-grained generated results.

## C   Importance of residual token embedding

To demonstrate the superior generalization of the residuals, we conduct comparative experiments. As shown in Tab. A1, compared to directly updating the class embedding parameters in a single text embedding, our approach, which involves updating the text encoder and calculating the average shift from the class to the specific subject based on a certain number of text templates, outperforms in both textual and visual similarity.

### C.1   Generated results of textual inversion

We observe from Fig. A8 that Textual Inversion [18] struggles with the generation of complex single subject and multiple subjects.

### C.2   About difference between residual token embedding and Textual Inversion

Textual inversion find a single word embedding (input of the text encoder) to represent user-provided subject. Different from textual inversion, we find the residual token embedding for each subject, and add these residuals to their corresponding token embedding (output of the text encoder). The "residual" actually refers to the ability to transform a subject of one category into a customized subject that we need, for example a "random dog" to the specific "customized dog". As shown in Fig. A9, we train model using method in textual inversion but learn a text embedding (output of the text encoder) and apply our layout guidance approach. We can see that simply employing Textual Inversion to learn a customized text embedding does not adequately fulfill the customized requirements. This approach fails to fully capture all the features of the reference subject, specially when dealing with multi-subject customization. Even with the utilization of our layout guidance method, better results cannot be achieved.

## D   Social impact and limitations

**Social impact.** While training individual large-scale diffusion models remains prohibitively expensive, advancements in fine-tuning techniques have enabled individual users to customize their own models. our approach empowers users to linearly combine their personalized single-subject models, generating high-quality images with multiple customized subjects while maintaining significant advantages in terms of computation and storage efficiency. Furthermore, there is a growing need for more reliable detection techniques to identify and mitigate the presence of fake data.

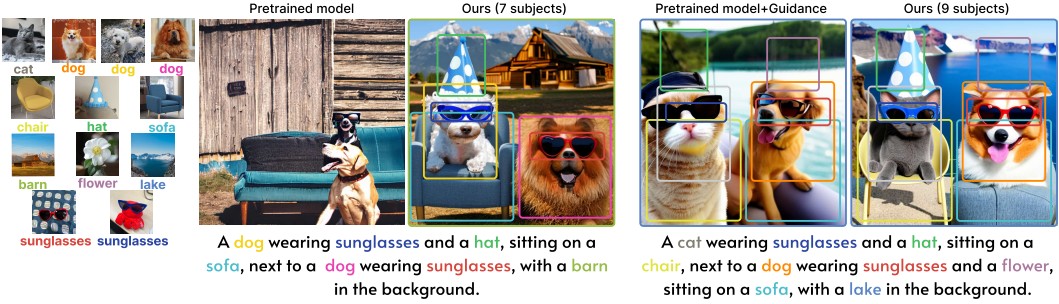

Figure A6: Generated results with more customized subjects. "Pre-trained model + Guidance" refers to apply our guidance approach to pre-trained model.

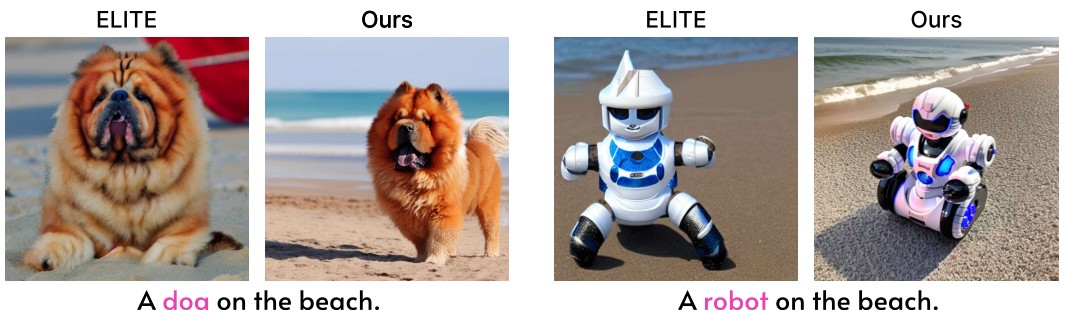

Figure A7: Comparisons with ELITE and our approach. Our method better preserves the identity of all subjects.

**Limitations.** our approach is limited by the inherent capabilities of the base model. Specifically, when it comes to combining more than six subjects, our approach may not be able to consistently generate satisfactory results. In order to achieve the desired generation results, the provided layout by the user needs to be roughly consistent with the textual description.

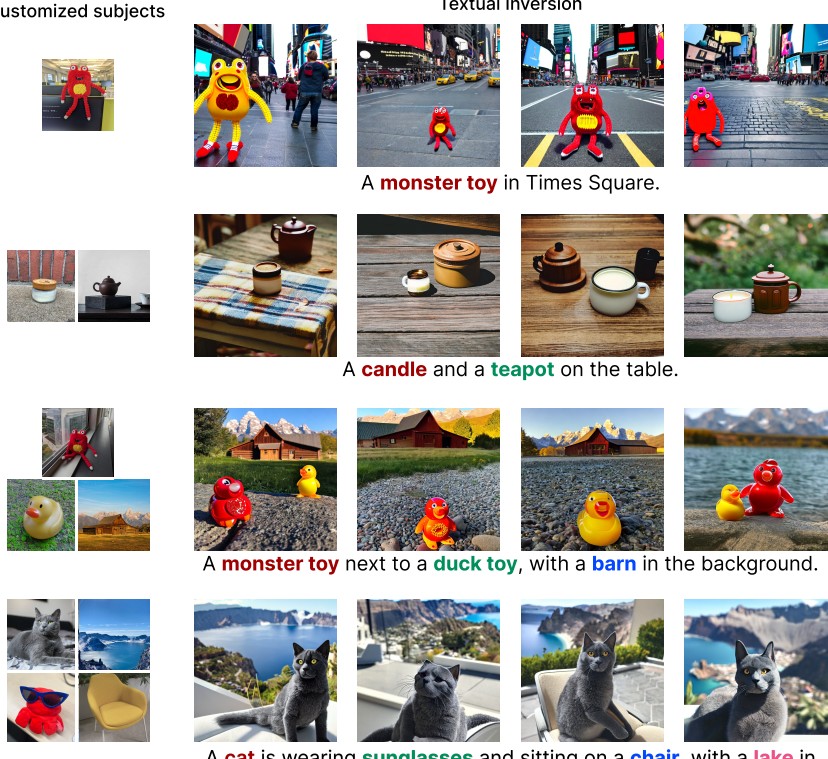

Figure A8: Generated results of Textual Inversion [18].

Table A1: **Quantitative comparisons** between our approach (learning a residual token embedding) and learning a token embedding directly.

|  | Single Subject | | Two Subjects | | Three Subjects | | Four Subjects | |
| --- | --- | --- | --- | --- | --- | --- | --- | --- |
|  | Text Alignment | Image Alignment | Text Alignment | Image Alignment | Text Alignment | Image Alignment | Text Alignment | Image Alignment |
| **Our approach** | 0.330 | 0.725 | 0.309 | 0.708 | 0.304 | 0.689 | 0.299 | 0.673 |
| **Token embedding** | 0.324 | 0.720 | 0.291 | 0.686 | 0.292 | 0.669 | 0.281 | 0.651 |

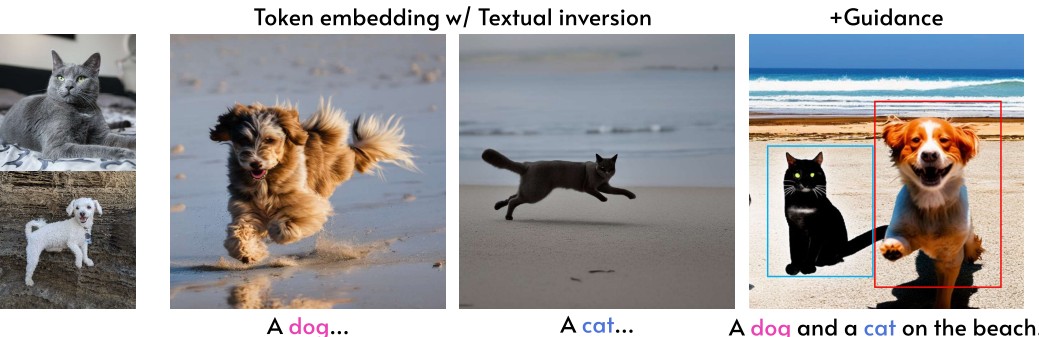

Figure A9: Visualization of learning embedding directly using Textual Inversion. "+Guidance" refers to applying our layout guidance.

