# OpenReview forum: "Customizable Image Synthesis with Multiple Subjects"
_NeurIPS.cc/2023/Conference — NeurIPS 2023 poster_

### Official Review · Reviewer_oYd1 · 2023-07-04

**Soundness:** 3 good
**Presentation:** 3 good
**Contribution:** 3 good
**Rating:** 6
**Confidence:** 4

**Summary:**

This paper aims to generate controllable images with multiple subjects as constraints. A residual token embedding is learned to shift the raw subject to the customized subject. A layout prior is further provided as the spatial guidance for subject arrangement. The experimental results demonstrate the effectiveness of the proposed method under a variety of settings.

**Strengths:**

The idea of using residual token embedding for a specific subject is a simple and effective way to generate customized subjects. The residuals and layout priors could be further utilized to adjust the attention for multi-subject generation without retraining. Compared to existing works, the proposed method enables the generation of a greater number of subjects for multi-subject generation.

**Weaknesses:**

1. The key contribution of this paper is the ability to generate a greater number of subjects compared to existing works. However, in general, the maximum number of subjects that this paper can deal with is 6. I was wondering about the comparison results that contain more than 6 subjects.
2. Different from existing works, this paper utilizes a predefined layout prior as the spatial guidance for multi-subject generation. Such result comparisons may be unfair, as the inputs of existing works do not contain the layout. Existing works that generate layouts from textual descriptions could be applied here to avoid predefining layouts.
3. I cannot find any quantitative results for the ablation study. In addition, how to determine the term that controls the relative weight of the text-embedding-preservation loss?
4. The authors state that they select cones as the baseline for customization with similar subjects, while Fig.4 shows the results of Dreambooth. I was wondering which baseline is actually used in this experiment.

**Questions:**

1. For an image with 6 or more subjects (e.g., Fig. 5), will the layout prior still be easy-to-obtain? It would be better to show all the predefined layouts in the figures.
2. How to determine the number of steps of guidance to generate satisfactory results?

**Limitations:**

Yes.

---

> ### Author Rebuttal · Authors · 2023-08-10
>
> ### **Author Response to Reviewer oYd1**
> We greatly appreciate all of your valuable suggestions, which play a pivotal role in enhancing the quality of our paper. Bellow we address all your concerns.
> #### **Q1: About generated results with more than 6 subjects.**
>
> **A1:** Thanks. As shown in Fig. 3 in our paper, When dealing with three subjects, our method has already demonstrated significant superiority over existing methods. As for more subjects, it is limited by the capabilities of the pre-trained model itself. As shown in Fig. R5 **in  the newly added PDF file**, more involved subjects usually decrease the final visual quality. we observe significantly more failure cases when generating five or more subjects. We will include this in the revision. Several recent works [1, 2, 3] point out that stable diffusion is a struggle in generating multiple subjects. A potential way to ease this issue may be to apply our method to a better text-conditioned diffusion model.
>
>
> #### **Q2: About avoiding predefining layout.**
>
> **A2:** Thanks. Generating layouts from textual descriptions automatically [4,5,6,7] makes the entire process more convenient, and we will also include relevant discussions in the revision.
>
> #### **Q3: About showing quantitative results for the ablation study.**
>
> **A3:** Thanks. Here we present the quantitative results of the ablation experiments for the two operations in our layout guidance: strength and weaken attention activations. The visualizations of these results correspond to Fig. 6 in our paper.
>
> | Average CLIP Image Similarity | Ours | Only Strength | Only Weaken |
> | :-: | :-: |:-: |:-: |
> | Single-subject | 0.7949 | 0.8081 | 0.7988 |
> | Two-subject    | 0.7075 | 0.6736 | 0.6568 |
>
> In addition, we conducted additional ablation experiments on the text preservation loss weight $\lambda$ we designed, and the quantitative results are as follows:
>
> | Average CLIP Image Similarity | $\lambda=0$ | $\lambda=0.5$ | $\lambda=1.0$ | $\lambda=2.0$ |
> | :-: | :-: |:-: |:-: | :-: |
> | Single-subject | 0.7900 | 0.7881 | **0.7949** | 0.7886 |
> | Two-subject    | 0.6196 | 0.6719 | **0.7075** | 0.6631 |
>
> As shown in the table, completely omitting the text-embedding preservation loss leads to a collapse in the performance of multi-concept customization. In practical implementation, we selected a loss weight of 1.0 for this regularization term.
>
> #### **Q4: About the baseline used in Figure 4.**
>
> **A4:** Thanks. Actually, we choose Cones as a baseline. In Fig. R11 **in the newly added PDF file**, we conduct a new experiment to compare the generation capabilities of DreamBooth, Cones, and our method in challenging cases.
>
> #### **Q5: Discussion about layout prior.**
>
> **A5:** In practice, users can simply select the customized subjects in the text prompt by clicking on them and then place and resize the bounding boxes accordingly. We show some examples of the bounding boxes we used in Fig. R3 **in the newly added PDF file** and will add all the bounding boxes we used in the revision.
>
> #### **Q6: About how to determine guidance steps.**
>
> **A6:** As shown in the two columns on the right in Figure 6 in our paper, for simple combinations like "mug + teapot," satisfactory results could be achieved with 30 steps of guidance. However, for more challenging combinations such as "cat + dog," 50 steps of guidance were required to achieve better attribute binding results.
>
> [1] Training-Free Structured Diffusion Guidance for Compositional Text-to-Image Synthesis. Feng *et al.* ICLR'23.
>
> [2] Attend-and-excite: Attention-based semantic guidance for text-to-image diffusion models. Chefer *et al.* SIGGRAPH'23.
>
> [3] MultiDiffusion: Fusing Diffusion Paths for Controlled Image Generation. Bar-Tal *et al.* ICML'23.
>
> [4] LayoutGPT: Compositional Visual Planning and Generation with Large Language Models. Feng *et al.* arXiv preprint arXiv:2305.15393.
>
> [5] Grounded Text-to-Image Synthesis with Attention Refocusing. Phung *et al.* arXiv preprint arXiv:2306.05427.
>
> [6] VisorGPT: Learning Visual Prior via Generative Pre-Training. Xie *et al.* arXiv preprint arXiv:2305.13777.
>
> [7] LLM-grounded Diffusion: Enhancing Prompt Understanding of Text-to-Image Diffusion Models with Large Language Models. Lian *et al.* arXiv preprint arXiv:2305.13655.

---

> > ### Comment · Reviewer_oYd1 · 2023-08-19
> >
> > I appreciate the authors' clarifications. My concerns have been addressed. I will increase my score to 6. Please involve the additional evaluations and discussions in the revised version.

---

> > > ### Author Response · Authors · 2023-08-19
> > >
> > > Dear reviewer oYd1, thank you very much for your affirmation. Your suggestions regarding the ablation experiments and predefining the layout through LLM contribute to enhancing the quality of our work. We will discuss and incorporate them in the revision. Once again, thank you for your patient response!

---

### Official Review · Reviewer_msmN · 2023-07-06

**Soundness:** 2 fair
**Presentation:** 3 good
**Contribution:** 3 good
**Rating:** 5
**Confidence:** 3

**Summary:**

This paper proposes a method to achieve customizable image generation with multiple subjects. To achieve the subject generation, it is done by learning a residual prediction for the subject tokens. To make multiple subject tokens combinable, it proposes a text-embedding preservation loss to make the embedding of the category from fine-tunes E and the freezed E to be the same expect for the subject. To ensure the subjects does not conflict and the spatial layouts, it additionally use bounding boxes to modulate the cross-attention when generate the outputs. The proposed method is compared with baselines such as DreamBooth, CustomDiffusion, Cones on the subjects previous works used. The results shows promising results on custimizable image generation with up to four or even more subjects.

**Strengths:**

- Proposed a efficient method to handle multiple instance of customization by learning a residual token from the base-category. This makes the customization succeeds without fine-tuning a large number of parameters.

- The text-embedding preservation loss is proposed to make multiple subject tokens combinable. I really like this simple yet effective design of the loss.

- Using layout to guide the generation process makes sense: it provides fine-grained control and avoid the conflicts between subjects.

- The paper is easy to follow and the presentation of the results is clear.

**Weaknesses:**


- Some important aspects of the method can be elaborated more clearly. For instance, how is the layout be taken for the model as the inputs? Is it also the input in the denoising steps or they are just used to find the corresponding area in the attention maps? Why do we need \ita(t) in Eq. 5? Are layouts used during the pretraining of the subject tokens?

- Although the method has been compared with previous works, some ablation studies and discussion of the limitation are missing. For instance, the quantitative evaluation of the ablation studies.

**Questions:**

- Ablation study for the proposed components such as text-embedding-preservation loss and the effects of the layout-guidance generation are missing.

- What's the limitations of the proposed methods beside scaling up to even more subjects?

- Typo: Line 245: Verift -> Verify.

**Limitations:**

The authors did not address the limitation and potential negative societal impact of their work. Subject generations can produce fake contents and might be dangerous if not used carefully. The authors should discuss this aspect of their work.

---

> ### Author Rebuttal · Authors · 2023-08-10
>
> ### **Author Response to Reviewer msmN**
> We greatly appreciate all of your valuable suggestions, which play a pivotal role in enhancing the quality of our paper. Bellow we address all your concerns.
> #### **Q1: More details about the layout guidance method.**
> **A1:** During inference, the layout provided by the user is not directly input into the model. The core of our proposed layout guidance method lies in editing the activation matrices of the cross-attention layers in each iteration of denoising to guide the model in generating the desired image. The layout serves as a reference for this editing operation.
>
> In Equation 5, $\eta(t)$ is a concave function designed as a noise scheduler referencing the pre-train model. It gradually decreases as $t$ decreases（from $T$ to 1). Its role is to weaken the intensity of layout guidance during the sampling process as time step $t$ decreases. In the sampling process of the diffusion model, as t decreases from $T$ to 1, the inference process can be understood as transitioning from determining high-level semantics to determining low-level semantics. The purpose of gradually decreasing $\eta$ with $t$ is to prevent excessive guidance from the layout when t approaches 0, which could result in noticeable disharmonious artifacts in the generated images.
>
> When training the residual token embedding for a specific subject, layout is not required. Only a set of multi-view reference images of this subject is needed.
>
> We will clearly add these details discription in the revision.
>
> #### **Q2: About supplementing ablation study.**
> **A2:** Thanks. Here, we present the quantitative results of three ablation studies. Firstly, we compare the approach of directly learning a text embedding like Textual Inversion with our design of learning residuals. The table below suggests that, without the residual design, the overall performance degrades given varying contexts.
>
> | Average CLIP Image Similarity | Ours | Learn Directly |
> | :-: | :-: |:-: |
> | Single-subject | **0.7949** | 0.6953 |
> | Two-subject    | **0.7075** | 0.6092 |
>
> Secondly, we present the quantitative results of the ablation experiments for the two operations in our layout guidance: strength and weaken attention activations. The visualizations of these results correspond to Fig. 6 in our paper.
>
> | Average CLIP Image Similarity | Ours | Only Strength | Only Weaken |
> | :-: | :-: |:-: |:-: |
> | Single-subject | 0.7949 | **0.8081** | 0.7988 |
> | Two-subject    | **0.7075** | 0.6736 | 0.6568 |
>
> Finally, we conducted additional ablation experiments on the text preservation loss weight $\lambda$ we designed, and the quantitative results are as follows:
>
> | Average CLIP Image Similarity | $\lambda=0$ | $\lambda=0.5$ | $\lambda=1.0$ | $\lambda=2.0$ |
> | :-: | :-: |:-: |:-: | :-: |
> | Single-subject | 0.7900 | 0.7881 | **0.7949** | 0.7886 |
> | Two-subject    | 0.6196 | 0.6719 | **0.7075** | 0.6631 |
>
> #### **Q3: About limitations of the proposed method.**
> **A3:** Besides scaling up to more subjects, our method cannot strengthen the interaction relationships between multiple subjects. The difficulty in generating complex interaction relationships of the pretrained model is inherited in our method. This issue is shared by all existing customized methods. As shown in Fig. R9 **in the newly added PDF file**, in the case of straightforward interaction relationships, such as "sit" and "wear," both our approach and the pretrained model achieve satisfactory generated results. However, for more intricate interaction relationships, such as "handshake," the performance of both our approach and the pretrained model falls short of expectations.

---

> > ### Comment · Reviewer_msmN · 2023-08-17
> >
> > Thanks for the additional ablation study on the model. I have read the responses and all my questions are resolved. I am willing to adjust the rating to 6.

---

> > > ### Author Response · Authors · 2023-08-18
> > >
> > > Dear reviewer msmN, thank you for your valuable suggestions on our work! Your suggestions regarding the ablation study are highly significant to our research, and we will incorporate the relevant experiments in the revision. We sincerely appreciate your recognition of our work and your willingness to raise the rating!

---

### Official Review · Reviewer_Nfe2 · 2023-07-06

**Soundness:** 2 fair
**Presentation:** 3 good
**Contribution:** 2 fair
**Rating:** 6
**Confidence:** 5

**Summary:**

This paper introduces a novel method for multi-subject, subject-driven text-to-image generation. The authors develop an efficient system that embeds single subject information effectively and seamlessly combines these separate subject embeddings to generate the final multi-subject image. The key concept is to learn a residual on top of the clip text-embedding for the subject token, thereby enriching it with subject-specific information. This residual is optimized so that it encapsulates specific information about the single subject without imposing further restrictions on the images. Another innovation is a test-time cross attention manipulation technique. Notably, most previous personalization approaches experience object neglect or attribute confusion as the number of subjects increases. This paper addresses these issues by strengthening and weakening certain regions of the cross-attention maps, guided by user-provided layouts. Both quantitative and qualitative results demonstrate superior performance compared to the DreamBooth, Custom-Diffusion, and Cone baselines

**Strengths:**

S1: The paper innovatively designs a preservation loss to optimize text embedding offset, enhancing class information preservation and reducing overfitting.

S2: The proposed layout manipulation effectively addresses object neglect and attribute confusion, enabling better scalability for larger numbers of subjects.

S3: The method significantly reduces storage and computational costs from exponential to linear, enhancing accessibility for multi-subject image generation.


**Weaknesses:**

W1: Despite the novel use of text embedding, the paper exhibits a limitation in this area. Without model fine-tuning, the images don't preserve subject detail as effectively as one might hope (e.g. see figure 1 white dog), which would be problematic when the subject details are important, e.g. when the subject is a human (which is not tested in this paper).

W2: The use of layout as guidance may also present some constraints. It relies heavily on user inputs, potentially limiting the system's flexibility and automation. Furthermore, it appears to be primarily suited to group photos, and may struggle with more diverse actions that involve intricate interactions.

W3: Although the paper presents an effective approach to reduce computation and storage costs, the process is still relatively expensive, particularly when compared to methods such as tuning an encoder [1] or ELLITE [2]. The requirement for fine-tuning limits the method's accessibility or widespread adoption.

W4: The paper's approach to the evaluation of multi-subject generation raises some concerns. The authors mentioned that "For multi-subject generation, we calculate the image similarity of the generated images and each target subject separately and finally calculate the mean value." A potentially more meaningful approach could be to perform object detection first, matching the resulting detections with the reference subjects. Without this, the similarity between single subject and multi subject image doesn't seem very meaningful.

[1] Gal, Rinon, et al. "Designing an encoder for fast personalization of text-to-image models." Siggraph 2023
[2] Wei, Yuxiang, et al. "Elite: Encoding visual concepts into textual embeddings for customized text-to-image generation." arXiv preprint arXiv:2302.13848 (2023).

**Questions:**

Here are a few minor comments and suggestions.

1. line 64 over -> out
2. EDIFF-I [1] uses similar cross-attention manipulation formulation for layout-to-image generation. See Section 4.3 of the paper. I suggest the authors to add some discussions.
3. figure 4 mismatch with the text description. Which baseline is used here? DreamBooth or Cone ?

[1] Balaji, Yogesh, et al. "ediffi: Text-to-image diffusion models with an ensemble of expert denoisers." arXiv preprint arXiv:2211.01324 (2022).

**Limitations:**

The authors partially address the limitations. Please find other suggestions in the weakness and question section.

---

> ### Author Rebuttal · Authors · 2023-08-10
>
> ### **Author Response to Reviewer Nfe2**
> We greatly appreciate all of your valuable suggestions, which play a pivotal role in enhancing the quality of our paper. Bellow we address all your concerns.
> #### **Q1: Ability to maintain subject detail.**
> **A1:** Thank you for the questions raised by the reviewer. As shown in Fig. R8 **in the newly added PDF file**, we show the generated results of the subjects robot and human, both of which attain comparable results. However, for subjects with a large number of intricate details, there is indeed a gap compared to model tuning methods. Finding better conditions to enhance the preservation of subject details is a promising direction for future improvement. However, for multi-subject generation, existing model tuning methods suffer from high training costs, subject disappearance, and attribute confusion. Our method exhibits superior performance.
> #### **Q2: Constraints of layout guidance.**
> **A2:** Actually, it is very easy to obtain a bounding box. Users can simply select the customized subjects in text prompt by clicking on them and then place and resize the bounding boxes accordingly. We do not make any corrections to the attention maps corresponding to relational words; thus, our ability to represent the interactions between subjects relies entirely on the pre-trained model. As shown in Fig. R9 **in the newly added PDF file**, in the case of straightforward interaction relationships, such as "sit" and "wear," both our approach and the pre-trained model achieve satisfactory generated results. However, for more intricate interaction relationships, such as "handshake," the performance of both our approach and the pretrained model falls short of expectations.
>  Improving the representation of interaction relationships between subjects is a common challenge faced by all existing customized generation methods. Currently, there are also some recent works [1, 2] exploring better ways to represent interaction relationships.
> #### **Q3: Computation comparisons with fast customized methods.**
> **A3:** We are very grateful for the reviewer's suggestion, and we will add discussion in the revision. Currently, some works [3, 4] focuses on achieving faster customized generation by pretraining an image encoder and using it to encode a reference image during generation. However, these methods share some common issues：
> - They require collecting data in advance to train the encoder, which can limit their generalization [1].
> - We compare our method with Elite in Fig. R10 **in the newly added PDF file**. Compared to methods that fine-tune for each individual subject, these approaches may exhibit difficulty in preserving fine details.
> - In addition, these methods can only encode one image at a time, they cannot customize multiple subjects simultaneously.
> #### **Q4: Quantitative evaluation with a detection-based metric.**
>
> **A4:** Great point! We sincerely appreciate the valuable suggestion. Our evaluation metric of multi-subject generation is inherited from Custom Diffusion and Cones. However, we firmly believe that the evaluation metric you mentioned is more reliable. Therefore, we conduct an evaluation and show the results in the table below.
>
> |Subjects | DreambBooth |Custom diffusion|Cones|Ours|
> | :-: |:-: |:-: |:-: |:-: |
> | 2| 0.7301 $\pm$ 0.0054 |0.7238 $\pm$ 0.0023 |0.7591 $\pm$ 0.0013 | **0.8107 $\pm$ 0.0004** |
> | 3| 0.6981 $\pm$ 0.0101 |0.7150 $\pm$ 0.0086 |0.7276 $\pm$ 0.0072 | **0.7752 $\pm$ 0.0031** |
> | 4| 0.6312 $\pm$ 0.0183 |0.6387 $\pm$ 0.0109 |0.6771 $\pm$ 0.0089 | **0.6987 $\pm$ 0.0042** |
>
> During the evaluation process, we apply GLIP [5] to detect the corresponding subjects in the generated images and crop the images accordingly. Then we calculate the CLIP image similarity for each customized concept and take the average. The results indicate that our method outperforms the baseline in terms of CLIP image similarity. Additionally, the evaluation shows that our method has a lower variance, indicating that it can generate all the required subjects in a more stable and consistent manner.
> #### **Q5: Discussion about difference from EDIFF-I.**
> **A5:** Different from existing methods [6] that strengthen the signal of the target subject within the layout-indicated area of the cross-attention map, we also propose to **weaken** the signal of irrelevant subjects in the same area. Such a design helps alleviate the issue of attribute mixing across different subjects, especially along with the number of subjects increasing. Fig. 6 in the submitted manuscript and the table below demonstrate the effectiveness of our weakening design both qualitatively and quantitatively.
>
> | CLIP Image Similarity | Ours | Only Strength | Only Weaken |
> | :-: | :-: |:-: |:-: |
> | Single-subject | 0.7949 | **0.8081** | 0.7988 |
> | Two-subject    | **0.7075** | 0.6736 | 0.6568 |
> | Two-subject (w/ Detection) | **0.8107 $\pm$ 0.0004** | 0.7508 $\pm$ 0.004 | 0.7993 $\pm$ 0.0006 |
> #### **Q6: The baseline used in Figure 4.**
> **A6:** We sincerely apologize for our oversight. Actually, we choose Cones as the baseline. In Fig. R11 **in the newly added PDF file**, we conduct a new experiment to compare the generation capabilities of DreamBooth, Cones, and our method in challenging cases.
>
> [1] ReVersion: Diffusion-Based Relation Inversion from Images. Huang *et al.* arXiv preprint arXiv:2303.13495.
>
> [2] ProSpect: Expanded Conditioning for the Personalization of Attribute-aware Image Generation. Zhang *et al.* arXiv preprint arXiv:2305.16225.
>
> [3] Designing an encoder for fast personalization of text-to-image models. Gal *et al.* SIGGRAPH'23.
>
> [4] Elite: Encoding visual concepts into textual embeddings for customized text-to-image generation. Wei *et al.* arXiv preprint arXiv:2302.13848.
>
> [5] Grounded language-image pre-training. Li *et al.* CVPR'22.
>
> [6] eDiff-I: Text-to-Image Diffusion Models with an Ensemble of Expert Denoisers. Balaji *et al.* arXiv preprint arXiv:2211.01324.

---

> > ### Comment · Reviewer_Nfe2 · 2023-08-17
> > **Response**
> >
> > Thank you for the reply. The new evaluation and comparison with EDIFF-I makes the paper stronger. I increase my score to 6. Regarding point 1,2,3, the limitations still hold so I suggest the authors to discuss this further in the revised version.

---

> > > ### Author Response · Authors · 2023-08-18
> > >
> > > Dear reviewer Nfe2, first and foremost, we want to extend our gratitude to you for your meticulous review and valuable feedback on our manuscript, particularly with regard to the evaluation metric of multi-subject generation. These suggestions are very important to refine our work! We will discuss the limitations in the revision. Once again, we want to express our sincere appreciation for your time, effort, and expertise.

---

### Official Review · Reviewer_QhyH · 2023-07-07

**Soundness:** 4 excellent
**Presentation:** 4 excellent
**Contribution:** 4 excellent
**Rating:** 6
**Confidence:** 4

**Summary:**

This paper studies how to efficiently represent a particular subject as well as how to appropriately compose different subjects. The author finds that the text embedding regarding the subject token can serve as a simple yet effective representation. To capture features of a specific subject, the author propose a text-embedding-preservation loss to learn a residual token embedding. Based on the residual embeddings, the author employ layout as the spatial guidance for subject arrangement into the attention maps. Both qualitative and quantitative experimental results demonstrate the superiority of the proposed method

**Strengths:**

a)	The paper is clear and easy to read.

b)	The proposed method shows superiority towards challenging cases, compared with existing methods.

c)	From both quantitative and qualitative results, the author conducted detailed experiments to analyze and demonstrate the effectiveness of the proposed method.


**Weaknesses:**

a)	What is the difference between residual token embedding with textual inversion? If a textual inversion is trained for each concept to obtain a token embedding, and then the layout is used for guidance, what are the results like?

b)	In Table 2, the author calculates the complexity of different methods. Regarding multiple subjects, what are the training time results between different methods?

c)	From the second row of figure 4, the color of the white puppy does not seem to have been well maintained.



**Questions:**

a) What is the difference between residual token embedding with textual inversion? If a textual inversion is trained for each concept to obtain a token embedding, and then the layout is used for guidance, what are the results like?

b) In Table 2, the author calculates the complexity of different methods. Regarding multiple subjects, what are the training time results between different methods?

c) From the second row of figure 4, the color of the white puppy does not seem to have been well maintained.

**Limitations:**

Please refer to the weaknesses part.

---

> ### Author Rebuttal · Authors · 2023-08-10
>
> ### **Author Response to Reviewer QhyH**
> We sincerely appreciate the affirmation from the reviewer for our work. It serves as a strong motivation for us! Bellow we address your concerns separately.
> #### **Q1: About difference between residual token embedding and Textual Inversion.**
> **A1:** Textual inversion find a single word embedding (input of the text encoder) to represent user-provided subject. Different from textual inversion，we find the residual token embedding for each subject, and add these residuals to their corresponding token/text embeddding (output of the text encoder). The "residual" actually refers to the ability to transform a subject of one category into a customized subject that we need, for example a "random dog" to the specific "customized dog". As shown in Fig. R6 **in the newly added PDF file**，we train model using method in textual inversion but learn a text embedding (output of the text encoder) and apply our layout guidance approach. We can see that simply employing Textual Inversion to learn a customized text embedding does not adequately fulfill the customized requirements. This approach fails to fully capture all the features of the reference subject, sepecially when dealing with multi-subject customization. Even with the utilization of our layout guidance method, better results cannot be achieved.
>
> #### **Q2: Training time results between different methods.**
> **A2:** We present the training time between different methods for single-subject generation in the table. All experiments are completed on a single 80G-A100.
>
> | Method | Textual inversion |DreamBooth |Custom diffusion|Cones|Ours|
> | :-: | :-: |:-: |:-: |:-: |:-: |
> | Training time| 30 minutes |15 minutes |10 minutes |10 minutes |20 minutes |
>
> It is important to note that our approach utilizes learned single-subject residual token embeddings, enabling seamless combinations without the need for retraining. This helps us avoid the exponential training costs associated with other methods. In contrast, other methods require additional storage space and training time for each new combination of subjects, and their training time increases linearly with the number of customized subjects.
>
> #### **Q3: Color of white puppy.**
> **A3:** Our training data is sourced from the official DreamBooth dataset. In Fig. R7 **in the newly added PDF file**, We present the training dataset for the "white puppy". This dataset contains instances of blurred and poorly lit images, which to some extent, influence the generated results.

---

> > ### Comment · Reviewer_QhyH · 2023-08-20
> > **Official Comment by QhyH**
> >
> > I would like to express my gratitude for the diligent efforts made by the authors in addressing my questions. The authors have addressed my concerns, and I will maintain my score unchanged.

---

> > > ### Author Response · Authors · 2023-08-21
> > >
> > > Thanks for your valuable suggestions and feedaback! we're also glad that we've addressed all your concerns. Lastly, we would like to express our gratitude for your time and insights.

---

### Official Review · Reviewer_X4Jp · 2023-07-07

**Soundness:** 3 good
**Presentation:** 4 excellent
**Contribution:** 4 excellent
**Rating:** 7
**Confidence:** 3

**Summary:**

This paper presents a new method for generating new images of any combination containing given objects. It combines individually learned single-subject residuals for multi-subject generation without retraining. The authors proposed to use text-based embedding to represent individual objects. Once a residual token was learned, they can then add these residuals to the embedding and adjust the attention maps based on a given layout. The proposed method outperforms other existing methods in all multi-subject customized tasks, especially in a three-subject and four-subject generation.

**Strengths:**

This is a timely paper that works on an interesting and challenging problem of controllable image synthesis based on the diffusion model. The authors proposed to compose multiple subjects by leveraging layout guidance. Such prior is simple and intuitive to use. The paper is well written and easy to follow. They performed extensive evaluation/ablation studies and outperforms other existing methods both quantitatively and qualitatively. They also conducted a user study to further evaluate the performance of their method.

**Weaknesses:**

This paper is a combination of many existing ideas, such as text embedding vector from zero-shot img2img paper, text-embedding-preservation loss, and cross-attention map. While the execution and presentation of these ideas are well done, I wes hoping for a more original contribution and unique solution to the problem. Overall, despite this, the paper still demonstrates impressive results and meets the standard for acceptance.

**Questions:**

1. Would the authors consider other image composition methods as baselines to this method? For example, CVPR 2023 paper titled "ObjectStitch: Object Compositing with Diffusion Model"?
2. Is there a way to apply more fine-grained control on the composition of multiple objects, for example, by applying a specific mask or changing the pose of each object in the new composition? does the proposed require non-overlapping boxes? It would be more informative to have examples of bounding box layouts on the side of each result.
2. What happened when the user-provided layouts are not consistent with the text description? Are there other potential failure cases for this method?

Typo: Line 245 Verift -> Verify


**Limitations:**

The authors address the limitations in supplementary material in that their method is limited by the inherent capabilities of the base model. The paper also discussed the potential societal impact of user-specific image generation. I think they are valid.

---

> ### Author Rebuttal · Authors · 2023-08-10
>
> ### **Author Response to Reviewer X4Jp**
> Thank you for your positive comments and valuable feedback on our work! We are excited and encouraged by your support!  Bellow we address your concern separately.
> #### **Q1: About the main contributions.**
> **A1:** Thanks. As you have pointed out, using diffusion models for customizable image synthesis is currently a crowded field and hence many similar techniques have been proposed. Compared to existing studies, our approach enjoys two original designs.
> - Regarding **subject representation**, we propose to learn a **residual** on top of the text embedding of a "base word" (*e.g.*, from "dog" to the "customized dog"). Unlike prior works that directly optimize the word embedding for the customized subject, our residual design helps the customization well blends with various context. Our motivation is that, assuming the text encoder could already adequately encode "dog" with different surroundings, the learned embedding shift could harmoniously blend "customized dog" with the same surrounding as well. The table below suggests that, without the residual design, the overall performance degrades given varying context.
> | Average CLIP Image Similarity | Ours | Learn Directly |
> | :-: | :-: |:-: |
> | Single-subject | 0.7949 | 0.6953 |
> | Two-subject    | 0.7075 | 0.6092 |
> - Regarding **subject arrangment**, we introduce layout as a very abstract and easy-to-obtain prior to guide the generation process. Different from existing methods that strengthen the signal of the target subject within the layout-indicated area of the cross-attention map, we also propose to **weaken** the signal of irrelevant subjects in the same area. Such a design helps alleviate the issue of attribute mixing across different subjects, especially along with the number of subjects increasing. Fig. 6 of the submitted manuscript and the table below demonstrate the effectiveness of our weakening design both qualitatively and quantitatively.
> | Average CLIP Image Similarity | Ours | Only Strength | Only Weaken |
> | :-: | :-: |:-: |:-: |
> | Single-subject | 0.7949 | 0.8081 | 0.7988 |
> | Two-subject    | 0.7075 | 0.6736 | 0.6568 |
>
> We will clearly explain our core contributions in the revision.
> #### **Q2: Comparison with image composition methods.**
> **A2:** Thanks. Since [1] mentioned by reviewer has no official open source, we select [2] as a baseline of image composition methods. In order to compare the effect of the baseline and our method more intuitively,  we use Paint by example to inpaint the image generated by our method with the reference image as input. We conduct visualization results in Fig. R1 **in the newly added PDF file**. Generated results of our method have better visual similarity, and we will provide more comparisons in the revision. Besides, We consider multi-subject customized generation is different with image composition in two a aspects.
> - The  purpose of multi-subject customized generation is to implant all user-provided subjects into the diffusion model; so that the model can generate various images of all subjects vividly guided by prompts. However, the purpose of image compositing is to insert an object into another image in a realistic way, without guidance from prompts.
> - Specifically, existing effective image composition methods require collecting data pairs and training an image encoder. During inference, they utilize the reference image as a condition to generate the image. In addition, to compose multiple subjects by the image composition methods, multiple iterations of inference are required, especially when there is an overlapping relationship among subjects. For example, in case "a cat wearing sunglasses sitting on a chair", inpainting "chair", "cat", and "sunglasses" in turn is necessary.
> #### **Q3: More fine-grained control.**
> **A3:** Actually, users have the flexibility to input a more specific mask, enabling them to achieve fine-grained control. In the paper, we choose bounding boxes due to their ease of acquisition. Experimental results in Fig. R2 **in the newly added PDF file** demonstrated that our method can use specific mask for guidance sampling. Our method supports overlapping boxes, and we demonstrate the bounding boxes we adopted in Fig. R3 **in the newly added PDF file**. Specifically, the bounding boxes for glasses and hat overlap with the bounding box for the dog. We will add more examples of the bounding boxes we used in the revision.
> #### **Q4: More failure cases.**
> **A4:** We add more failure cases **in the newly added PDF file**. Specifically, in Fig. R4 we show a case that user-provided layouts is contradict to the text description. In Fig. R5, we show that our method is limited by the performance of the pretrained model when it comes to customizing the generation of more than 6 subjects. When dealing with 7 concepts, the performance of the pretrained model deteriorates. In our generated results, details of certain subjects are not fully preserved. With the generation of 9 subjects, the combination of the pretrained model and our guidance method contributes to some improvement in the generation quality. However, at this point, both the pretrained model and our method exhibit some subject disappearance, and our method fails to maintain the identity of all subjects under such circumstances.
>
> [1] ObjectStitch: Object compositing with diffusion model. Song *et al.* CVPR'23.
>
> [2] Paint by example: Exemplar-based image editing with diffusion models. Yang *et al.* CVPR'23.

---

### Author Rebuttal · Authors · 2023-08-10

## Author Response to All:
Dear reviewers,

We thank all reviewers for their time and efforts in reviewing our paper. These constructive reviews can bring multiple improvements for our manuscript. We are encouraged that the reviewers appreciate our method, including
* innovatively designs [Reviewer Nfe2]
* a simple and effective method [Reviewer oYd1]
* outperform prior methods both quantitatively and qualitatively [Reviewer X4Jp,QhyH]
* well written and easy to follow [Reviewer X4Jp,QhyH,msmN]
* efficient storage and computational cost [Reviewer Nfe2,msmN]


We also have made diligent efforts to address all the concerns raised point by point. In this rebuttal, we have incorporated some new figures to more effectively address the concerns. Kindly review the newly uploaded one-page PDF.
* Figure R1 compares our method with Paint by Example. Our method can better preserve the identity of all customizable objects. [Reviewer X4Jp, Q2]
* Figure R2 gives generated images using specific semantic mask. We use a more specific semantic mask as a prior to get more fine-grained customizable objects. [Reviewer X4Jp, Q3]
* Figure R3 gives generated results using overlapping boxes. [Reviewer X4Jp, Q3; oYd1, Q5]
* Figure R4 gives generated results with inconsistent layouts and prompts. [Reviewer X4Jp, Q4]
* Figure R5 gives generated results with more customized subjects. [Reviewer X4Jp, Q5; oYd1, Q1]
* Figure R6 shows the effect of combining residual token embedding with textual inversion. [Reviewer QhyH, Q1]
* Figure R7 includes the examples of white puppy. [Reviewer QhyH, Q3]
* Figure R8 shows generated results of robot and human. [Reviewer Nfe2, Q1]
* Figure R9 shows generated results with interactions. [Reviewer Nfe2, Q2; msmN, Q3]
* Figure R10 compares our method with ELITE. [Reviewer Nfe2, Q3]
* Figure R11 compares our method with Dreambooth and Cones on some challenging cases. [Reviewer Nfe2, Q6]

We are open to discussions and addressing any issues from reviewers. Your constructive comments can further help us to improve our method.

Sincerely yours,

Authors

---

### Decision · Program_Chairs · 2023-09-21

**Decision:**

Accept (poster)

**Comment:**

This paper got positive recommendations: one "accept", three "weak accept", and one "borderline accept".

The reviews pointed out issues and concerns about the main contribution and possible comparisons with other methods. The authors' rebuttal emphasized the advantages of adopting the residual design and weakening the signal of irrelevant subjects. The rebuttal also provided additional results, including training time, quantitative evaluation with another metric, and ablation studies.

The rebuttal seemed to successfully address the reviewers' questions and main concerns. After the rebuttal, both **Reviewer Nfe2** and **Reviewer oYd1** increased their rating to "weak accept". **Reviewer msmN** also mentioned increasing the score to "weak accept" in the discussion but somehow did not make it in the final rating.

The AC agrees with the reviewers' unanimous recommendations for accepting the paper. The AC recommends that the authors incorporate the reviewers' feedback and the additional results into the final version.